# Methodological Development of a Test for Salivary Proteome Analysis Useful in Lung Cancer Screening

**DOI:** 10.3390/ijms26167924

**Published:** 2025-08-16

**Authors:** Leonarda Barra, Elena Carestia, Giulia Ferri, Mohammad Kazemi, Massoumeh Ramahi, Uditanshu Priyadarshi, Velia Di Resta, Fabrizio Di Giuseppe, Renata Ciccarelli, Achille Lococo, Stefania Angelucci

**Affiliations:** 1General and Thoracic Surgery Unit-Casa di Cura Synergo-Pierangeli, Piazza L. Pierangeli 1, 65100 Pescara, Italy; nadiabarra@hotmail.it (L.B.); diresta.velia@gmail.com (V.D.R.); chirurgiatoracica.pierangeli@grupposynergo.com (A.L.); 2Center for Advanced Studies and Technologies (CAST), ‘G d’Annunzio’ University of Chieti-Pescara, Via 9, Luigi Polacchi 13, 66100 Chieti, Italy; carestiaelena@gmail.com (E.C.); giulia.ferri1@outlook.it (G.F.); mohammad.kazemi@phd.unich.it (M.K.); massoumeh.ramahi@phd.unich.it (M.R.); uditanshu.priyadarshi@phd.unich.it (U.P.); fabrizio.digiuseppe@unich.it (F.D.G.); stefania.angelucci@unich.it (S.A.); 3Department of Aging Medicine and Sciences (DMSI), ‘G d’Annunzio’ University of Chieti-Pescara, Via Vestini 31, 66100 Chieti, Italy; 4Department of Sciences, G d’Annunzio’ University of Chieti-Pescara, Via Vestini 31, 66100 Chieti, Italy

**Keywords:** cancer lung, saliva, methodological approach, protein assay, two-dimensional electrophoresis, proteomic analysis, cancer biomarkers

## Abstract

Early diagnosis of lung cancer, essential for reducing its high mortality rate, is currently challenging, partly due to the lack of specific biomarkers. Here, we attempted to develop a noninvasive and potentially sensitive screening method based on the proteomic analysis of unstimulated and stimulated saliva samples, collected by passive drooling and salivary swabs, respectively, from healthy heavy smokers enrolled in a nonprofit screening project. Protein content analyzed before and after sample cryopreservation for various periods and the associated two-dimensional electrophoresis revealed that protein extraction after short-term cryopreservation prevented the loss of detectable proteins. Mass spectrometric analysis of these electrophoretically resolved proteins revealed the presence of salivary proteins whose levels may be dysregulated in various types of lung cancer. Finally, in pilot experiments conducted on stimulated saliva from a patient with a lung cancer nodule, we detected altered content or selective presence of proteins involved in lung carcinogenesis, such as serpin B3 or the proteins S100A14 and aldoketoreductase-A1, respectively. While acknowledging that these findings require further validation, we believe that the use of saliva and related proteomic analyses may contribute to the identification of potential early lung cancer biomarkers, which could hopefully improve clinical management of the tumor and patient survival.

## 1. Introduction

Lung adenocarcinoma is a tumor characterized by pulmonary nodules that can be classified as solid or nonsolid lesions. Solid lesions, surrounded by a well-defined capsule, are easily detected by computed tomography (CT). In contrast, nonsolid lesions, classified as pure, if completely nonsolid, or mixed, if they also have a solid component, appear on CT with a ground-glass opacity (GGO). This nonspecific finding is indicative of a variety of lung pathological conditions that require close follow-up every three months to monitor disease progression [1,2]. These lesions are associated with a poorer prognosis.

However, early diagnosis of lung adenocarcinoma is still challenging, making it one of the cancers with the highest mortality [3]. In the early stages, the disease is often asymptomatic, which prevents effective treatment that can ensure complete remission. Thus, while the five-year survival rate is very high if the cancer is diagnosed early, it decreases dramatically as the disease progresses [4,5].

Currently, there are no noninvasive diagnostic tests for lung adenocarcinoma. Low-dose computed tomography (LDCT) is the first-choice test, as it reduces radiation exposure compared to traditional CT [6]. However, it has a high rate of false positives, forcing patients to repeat the test several times to confirm or exclude the presence of the tumor. Furthermore, CT alone cannot determine with certainty the benign or malignant nature of smaller lesions, making periodic checks necessary to monitor their growth and evolution. Consequently, only 15% of lung adenocarcinomas are diagnosed at the early stages, while in most cases, symptoms appear only in the metastatic phase, compromising the effectiveness of treatment. Furthermore, this neoplasm mainly affects elderly people, often heavy smokers with smoking- and age-related diseases. In these patients, chemotherapy can be more harmful than beneficial [1,7,8]. Therefore, it is easy to understand why it is necessary to develop new diagnostic tests that allow for early diagnosis that are not harmful to the patient [9].

In the last decade, liquid biopsy has also been widely used for early diagnosis, prognosis, and disease monitoring in lung cancer (LC), although its routine clinical application still requires technical improvements and more dedicated studies [10,11]. Liquid biopsy has been performed using different types of biological fluids, mainly blood/serum and urine, but also sputum, while the evaluation of saliva is more recent. Saliva appears to have several advantages over other fluids studied so far, but it has some disadvantages. Among the advantages, easy accessibility and the need for minimally invasive methods to collect this fluid at relatively low costs compared to techniques currently available for other fluids have been highlighted. Furthermore, saliva mirrors the composition of blood due to physiological exchange mechanisms between the salivary glands and the systemic circulation [12]. More importantly, it also contains numerous biomolecules originating not only from the salivary glands but also from the nasal cavities and lower respiratory tract [13]. Therefore, it is not surprising if systemic diseases, including tumors such as LC, influence the salivary gland function as well as saliva composition [14,15]. However, some challenges remain for routine diagnostic saliva use, such as the presence of (i) highly abundant proteins (HAPs) [16,17] that can mask the presence of less abundant proteins (approximately less than 2% of total salivary proteins) that often include tumor biomarkers; (ii) contaminants, such as microorganisms and plasma proteins, which can alter results, especially in the presence of inflammatory foci that increase the amount of gingival crevicular fluid in saliva; and (iii) high interindividual variability in salivary protein composition, which could make it more difficult to identify biomarkers exclusively associated with the disease.

However, given the growing interest in identifying salivary biomarkers in LC, we decided to analyze the salivary proteome in an attempt to identify potential tumor biomarkers. Of course, we knew that achieving this goal required applying a new methodological approach and performing a series of experiments to highlight the technical features necessary to avoid invalidating future results.

We therefore initiated a methodological study, in which we collected and analyzed saliva samples from a large number of heavy smokers enrolled in a population-based health screening program promoted by the regional nonprofit organization LASMOT (Abruzzo League for the Support of Patients with Thoracic Tumors). The subjects were between 55 and 85 years old, had not quit smoking for more than 55 years, and did not have suspicious lung nodules. Only as an example, we also reported some preliminary results obtained by applying the method described here to saliva obtained from a patient with a CT-positive nodule. Our hope is that this approach will lead to the future identification of potential salivary biomarkers useful for the early diagnosis of lung diseases, paving the way for the development of inexpensive and minimally invasive diagnostic tests that allow for the targeted screening of individuals at risk for LC, such as smokers.

## 2. Results

### 2.1. Saliva Collection and First Phase of the Experimental Protocol

Saliva samples were collected from subjects enrolled in the aforementioned research program using two methods, described in detail in the Section 4. Thus, we obtained unstimulated saliva samples, i.e., the saliva normally present in the mouth, by passive drooling while the use of devices known as Salivettes promoted saliva production directly from the salivary glands, resulting in protein-rich samples. Proteins were then extracted and measured from both types of collected samples, and their profiles were analyzed using two-dimensional electrophoresis (2DE) combined with mass spectrometry (MS). Using these techniques, we sought to standardize the analytical procedures as much as possible in order to conduct a comparative analysis between salivary protein maps from different subjects.

We first examined the timing of protein extraction/measurement from saliva samples in relation to their storage at −80 °C for various periods. This step was important both to identify the presence or absence of contaminants that could alter the electrophoretic protein profile of the analyzed saliva and to ensure sufficient protein yield for qualitative and quantitative proteomic analysis. In particular:In unstimulated saliva samples, we evaluated the effect of the length of the sample cryopreservation period at −80 °C (2, 6, and 12 months) on the quantity and quality of proteins extracted at the end of the cryopreservation period or before cryopreservation;Based on the results obtained in the previous point, in stimulated saliva samples, i.e., those collected with the Salivette device, we evaluated the effect of shorter sample cryopreservation periods (1, 15, and 30 days) on the protein yield and electrophoretic profile. The evaluation of sample protein concentration was performed only at the end of the cryopreservation time.

#### 2.1.1. Evaluation of Protein Stability in Unstimulated Saliva Samples in Relation to Protein Extraction Times and Cryopreservation Duration

Nine unstimulated saliva samples belonging to subjects who tested negative on CT scans, i.e., controls, were randomly selected; proteins were extracted from each sample immediately after collection (time 0, T_0_) and after 2, 6, and 12 months from collection (three samples for each cryopreservation period; see Table 1). While the mean unit concentration of proteins extracted from saliva samples at the end of 2 months of cryopreservation was not significantly different from that measured at T_0_, the mean values of the samples after 6 and 12 months of cryopreservation were significantly lower than those measured at T_0_.

These data, in addition to showing a certain variability in the protein content among the chosen samples, indicated an acceptable stability of the salivary proteins only after a relatively short cryopreservation period (2 months). To confirm this aspect, we compared the 2DE maps obtained from some of the same saliva samples cryopreserved for different periods and whose proteins were extracted after cryopreservation. The 2DE analysis was carried out in technical triplicate on homogeneous gels (12%) (Figure 1). The electrophoretic run for the samples identified as 83C, 51C, and 11C resolved 173 ± 28, 102 ± 16, and 541 ± 66 total spots, respectively, distributed in a pH range of 4–7.

The comparison between these three biological replicates highlighted a low degree of similarity between the numbers of common spots and related percentages calculated vs. the total spot number revealed in the 83C and 51C samples and those found in the sample 11C cryopreserved for 12 months (Table 2). The significant spot increase pointed out in this sample, combined with a decrease in protein content measured at the end of the cryopreservation period, suggested the occurrence of a long-term denaturation of the protein components. This event, probably due to the activity of proteins/contaminants present in the unstimulated saliva samples, would lead to the formation of peptides/protein fragments that can be detected by electrophoresis while escaping accurate protein determination.

We then assessed whether protein extraction before sample cryopreservation could improve its stability. To this end, we examined the unstimulated 11C saliva sample, which in the previous series of experiments had shown the highest number of spots in the 2DE maps. Proteins were extracted immediately after sample collection and before cryopreservation. However, with this procedure, the mean protein concentration of this sample was significantly reduced even after only 2 months of cryopreservation following the extraction process (Table 3). This result suggests that denaturation processes still occurred, compromising the correct assessment of protein quantities, at least with the method used in our laboratory.

Again, the instability of salivary proteins extracted and then cryopreserved was confirmed through comparative analysis between the 2DE maps of the sample performed in technical triplicate for each time interval considered (i.e., 2, 6, and 12 months of cryopreservation from protein extraction) (Figure 2). Despite a satisfactory degree of reproducibility of the salivary proteomic profile on each technical triplicate, the total number of visible spots of the 11C sample after 2 and 6 months of cryopreservation was higher than that measured for the other two samples shown in Figure 1. Conversely, a decrease in the total number of visible spots was observed at 12 months compared to the value previously obtained for the same sample from which the proteins had been extracted at the end of the cryopreservation period. However, the protein concentration assessed after the same cryopreservation period was similar for the two samples.

Furthermore, the degree of gel similarity for the sample 11C indicated a correspondence of only 40% and 56% between the total proteome relative to 2 and 6 months, respectively, and that referred to 12 months of cryopreservation after extraction (Table 4).

In summary, although protein extraction before cryopreservation appeared to reduce the formation of peptide fragments identified by 2DE after long-term cryopreservation (12 months), the variation in protein concentration and total number of spots persisted at the end of the other cryopreservation period, likely due to degradation of the extracted proteins, which were still enriched in lysed proteins free in solution and easily susceptible to degradation. This aspect was contradictory and likely required further investigation on a larger number of saliva samples.

However, since early protein extraction from saliva samples offered no advantages over protein extraction after cryopreservation, we decided to continue our experiments by extracting salivary proteins after their storage at −80 °C.

#### 2.1.2. Evaluation of Protein Stability in Stimulated Saliva Samples in Relation to a Shorter Duration of Their Cryopreservation

In this set of experiments, we reduced the sample cryopreservation period, based on the results obtained from the previous phase. Thus, we analyzed nine randomly selected samples from those collected by Salivette. As reported in Table 5, the protein concentration of stimulated salivary samples showed a significant reduction in mean unit concentration equal to 1.751 ± 0.189 µg/µL after shorter cryopreservation periods (1–30 days), compared to the value of samples collected in the absence of stimulation and cryopreserved for longer periods (2–12 months) before protein extraction (as reported in Table 1).

However, despite the reduced sample cryopreservation duration, a significant decrease in protein concentration was still observed in the samples cryopreserved for longer periods (15 and 30 days) compared to that detected in three samples 1 day after cryopreservation. Although the use of Salivette ensured filtration of the saliva samples with removal of contaminants, this did not prevent protein degradation, which likely favors aggregation and precipitation processes in the salivary sample even over a short interval (15–30 days) and the loss of identifiable proteins. This finding was confirmed by the electrophoretic analysis of the samples. Indeed, the 2D run on 12% homogeneous gels for the samples collected by filtration with Salivette in technical and biological replicates resolved a decreasing number of detected spots equal to 496 ± 50, 466 ± 58, and 299 ± 32 protein spots, at 1, 15 and 30 days, respectively, distributed in a pH range of 4–7 (Figure 3). The degree of similarity was also progressively and significantly decreased during cryopreservation up to 30 days (Table 6). Nonetheless, it should be emphasized that the total spot number resolved by 2DE from stimulated saliva samples was higher than that obtained using samples from unstimulated saliva.

These results confirmed that, despite cryopreservation, protein denaturation processes occur, leading, over time, to a progressive loss of the total number of spots. Therefore, based on all the findings obtained so far, we decided to continue our study on unstimulated and stimulated saliva samples, which should be analyzed as soon as possible, i.e., within 1 week after their collection/cryopreservation. In any case, proteins were extracted and their concentration assayed at the end of the cryopreservation period, as this procedure also ensured faster preservation of the samples.

### 2.2. Second Phase of the Experimental Protocol: Comparison Between 2DE Maps of Unstimulated and Stimulated Saliva Samples Cryopreserved for 1 Week and Assayed for Protein Concentration at the End of the Cryopreservation Period

In this second phase, we used a gel-based approach to define the salivome by loading 10 µg and 25 µg of proteins for the unstimulated samples and the stimulated ones (Salivette), respectively. The electrophoretic run, performed on homogeneous gels (12%) at a pH 4–7 gradient and on 7 cm long gel strips, resolved a greater number of spots in the stimulated saliva that in unstimulated samples.

In Figure 4, we reported the image of one of the replicates of stimulated samples as a reference condition, characterized by a total number of well-resolved spots equal to 495, in comparison with that of an unstimulated sample, in which 386 spots were identified. The comparative analysis of the three technical replicates for each condition provided a good degree of intra-class similarity equal to 85%.

However, there was a difference in the number of resolved protein spots related to each condition (stimulated vs. non-stimulated saliva). As reported in Table 7, the matching analysis indicated a significantly higher number (*p* < 0.05, Student’s *t* test) of total spots equal to an average value of 453.3 ± 50.62 in the three stimulated samples compared to that of the three non-stimulated ones (329.3 ± 49.4), attributable to the collection method based on the Salivette device. As well, the spot matching expressed as a % value calculated versus the highest spot number identified in 2DE gel (i.e., 153S sample) showed a greater degree of reproducibility of the 2DE maps of the stimulated saliva (average percentage value around about 75%) as compared to the percentage in non-stimulated saliva (around 40%), as 399.7 ± 83.7 and 200 ± 67.4 common spots were revealed in stimulated and unstimulated saliva sample maps, respectively, while a higher number of unmatched proteins were detected in saliva in the absence of stimulation, equal to an average value of 129.3 ± 49.1 spots, compared to the stimulated samples (53.7 ± 51.2 spots).

Altogether, these data suggest a significantly higher sensitivity and specificity of the gel-based approach for stimulated samples obtained by Salivette that could be correlated to the filtration effect capable of producing more specific information about saliva proteome.

### 2.3. Effect of the Removal of High Abundant Proteins (HAPs) from the Saliva Samples

Like other biological fluids (urine, serum, and whole blood, plasma), saliva is characterized by the presence of highly expressed proteins, the HAPs; consequently, protein separation and their complete identification by 2DE is difficult due to the poor visualization of the molecular components with the lowest expression levels, the low-abundance proteins (LAPs). In pivotal studies conducted in our laboratory, in order to fully characterize the salivary profile, the protein equalization technique was adopted, based on the use of a suitable device known as Proteominer. This is a protein enrichment kit capable of optimizing the samples under examination by excluding interference with proteins normally present in abundance in biological fluids [18,19], but it has never been applied to saliva. We thought that this technical step could facilitate the identification of some inducible proteins specific for the type of sample under study in the screening of smoking-dependent lung cancer. However, a comparative analysis of salivary fluids treated and untreated with the Proteominer (Figure 5) showed that, although the amount of some LAPs, mostly visible in the lower part of the gel, increased, most HAPs were not significantly removed. Therefore, we no longer used this device.

### 2.4. Building a Reference Proteome Using Unstimulated and Stimulated Saliva Samples

In this phase of the study, we used saliva samples collected by drooling or Salivettes, with a one-week sample storage period, and protein extraction from these samples was performed after cryopreservation. We analyzed the saliva proteome obtained from 171 healthy subjects (i.e., without suspicious lung nodules on CT) enrolled in the LASMOT screening project, all heavy smokers and aged between 55 and 85. Comparative analysis of 2DE gels (24 cm gel strips) obtained from both types of saliva samples facilitated the identification of a greater number of proteins than gels obtained from each type of saliva sample (Figure 6). Indeed, while the mean number of identifiable protein spots was 895 ± 21 in unstimulated saliva and 979 ± 35 in stimulated saliva, the 2DE map constructed from the aforementioned comparison was characterized by a total number of 1180 protein spots distributed along a pH gradient between 4 and 7 and with Mr between 150 and 14 kDa. The increase was due to the inclusion of proteins identified by MS both as common to the two saliva types and as differently/exclusively expressed in stimulated saliva samples compared to the unstimulated ones. The identification of shared proteins was confirmed by scatter plot analysis, which analyzed interclass-sample variability by measuring the correlation coefficient value of all common spots compared (Appendix A). This analysis allows for a statistical assessment of the degree of similarity between 2DE gels, derived from the number of spots shared between saliva samples, while reproducibility is based on the correlation coefficient value, which in our case, was close to 1, being equal to 0.7.

Based on all the elements reported above, we constructed a master gel by comparing reference gels, which in turn were obtained from the previous comparison of 2DE gels run in triplicate with the proteins from each saliva sample. It was representative of smoking and CT-negative subjects, defined as controls (C) (Figure 6) and included proteins only partially exclusive to the biological compartment under examination, i.e., saliva.

These proteins included both HAPs, such as plasma albumin; immunoglobulin receptors; amylase (50–60%); the k chains of IgG; the PrP; and proteins rich in proline, such as Mucins, and LAPs, with a molecular weight equal to or below 20 kDa, such as variants of Cystatins (40–50%), which represent the so-called Depth Salivary Proteome, a group in which it should be possible to identify potential biomarkers.

Among them, we performed a spot selection based on the percentage intensity value with *p* < 0.05 and an expression level ≥ 2. From the respective preparative gels produced for each sample, the spots of interest were excised and, after tryptic digestion, analyzed by MS. In this way, we identified proteins in common in both stimulated and unstimulated saliva samples and labeled them as matched (M). These proteins showed a very similar amount in both sample types. However, other proteins resolved by 2DE were identified only in Salivette-collected saliva samples by MS analysis. They were up- or down-regulated as compared to those evaluated in unstimulated saliva and were labeled as D (differently expressed). Table 8 reports all selected proteins, while Figure 7 shows the statistically significant differences in the amount of those included in the lower part of Table 8.

The peptide sequences of all proteins reported in Table 8, whether common or differentially expressed, were validated using LIFT technology (described in the Section 4). With this method, we obtained parental ion masses from PMF spectra and the high Tof-Tof Score values unequivocally confirmed the unique peptide sequences (reported in red) for each protein (Appendix A).

Looking at Table 8, in the first group of proteins reported there, some of them were not specific for saliva such as interferon-alpha-1/13 (IFNA1) and alpha enolase isoform 3 (ENOB). IFNA1 is present in the extracellular matrix usually produced by macrophages with antiviral activities, while ENOB is a glycolytic metalloenzyme that catalyzes the dehydration of 2-phospho-D-glycerate (PGA) to phosphoenolpyruvate (PEP). It has been mainly found in striated muscles and may also bind to cytoskeletal and chromatin structures playing a role in transcriptional processes [20]. Furthermore, two isoforms of the protein complexes known as PIGR (polymeric immunoglobulin receptor) and the heat shock protein 70 kDa protein 1A (HS71A) were detected in both types of saliva samples. The two first proteins are IgA and IgM binding receptors (PIGR) usually produced in the basolateral portion of epithelial cells and then transferred on the apical portion of the same cells to transport IgA and IgM. Secreted IgA and IgM have vital roles in the immune response of oral mucosa to pathogenic infections [21]. Likewise, HS71A, a member of the heat shock protein 70 family, is implicated in a wide variety of cellular processes, including protection of the proteome from stress, folding and transport of newly synthesized polypeptides, activation of proteolysis of misfolded proteins, and the formation and dissociation of protein complexes [22]. It may also serve as a host cell receptor for virus entry and its levels are increased in saliva from obese pregnant women with periodontitis [23,24].

Differently, other protein seems to be more specific for saliva. One of these is S100A9, which belongs to the Annexin family known as S100 proteins, with a PM around 12 kDa, expressed in vertebrates. Their function is to bind calcium ions and to interact with other isoforms to form homo- and hetero-dimeric complexes, with indirect antimicrobial activity, since they are capable of acting as chelating substances towards transition metals (manganese, iron, and zinc), useful for the proliferation of pathogenic microorganisms [25,26]. Of interest is the presence of two isoforms of type 2 cystatin, S and N cystatins, among the common proteins detected in our samples. They are mainly secreted from parotid and submandibular glands and therefore found in human saliva, where they show antimicrobial activity against bacterial pathogens. Additionally, they may play an immunomodulatory role [27]. The isoform of carbonic anhydrase (CAH6) is also produced and secreted by all salivary glands; it is a metalloenzyme containing zinc and responsible for oral pH homeostasis; it prevents the demineralization of dental tissue by binding to the enamel film, where it catalyzes the conversion of bicarbonate, limiting the bone remodeling process. Together with proline-rich proteins and cystatins, it has been related to bitter taste perception [28]. Finally, in this group, particular attention should be paid to alpha amylase. It was present as two different isoforms, i.e., salivary alpha-amylase type 1 (AMY1A) and pancreatic alpha amylase (AMYP). Indeed, alpha-amylase is the most abundant isoform in saliva, produced mainly by the salivary glands, but it is also released by the pancreas.

Furthermore, we detected some proteins (labelled as D in Table 8) exclusively or differently expressed in stimulated saliva samples as compared to the unstimulated ones. Among those present only in stimulated saliva samples, there was leukocyte elastase inhibitor (ILEU), a 43 kDa protein. It is a physiological inhibitor of the proteases that are important in immune defense and was shown to be up-regulated in diabetic patients with periodontitis, suggesting that it might be responsible for lowering the immune defense system, leading to destruction of the periodontium; matrix proteins; and subsequently, the alveolar bone [29]. Another protein of this group was actin-related protein 3 (ARP3). It belongs to the cytoskeletal protein family, together with two other overexpressed proteins, i.e., constitutive type 1 (cytoplasmic) isoforms of actin (ACTB and ACTG). They are highly conserved proteins involved in cell motility, structure, integrity, and intercellular signaling. While they are not major salivary proteins, they can be found in saliva due to the presence of cells in the oral cavity that contain these proteins, such as epithelial cells. In particular, Arp3 is an ATP-binding component of the Arp2/3 complex which can build an actin network favoring the movement of immune cells against bacterial infection [29]. Accordingly, when the Arp3 production is decreased, the binding of ATP to produce the Arp2/3 complex is also reduced, making the host more susceptible to infection.

Furter proteins present only in stimulated saliva samples were 3′5′ exoribonuclease 1 (ERI1) and serpin E3 (SERP3). 3′5′ Exoribonuclease 1 (ERI1) is an enzyme that plays a role in RNA processing and turnover. Although it is conserved across many species, its precise role and impact in human health are still being investigated, and there is no direct link between ERI1 and saliva in the literature. Regarding the other protein, SERP3, it belongs to the superfamily of protease inhibitors that prevent excessive proteolysis. Its role has also not yet been well defined; it appears to play a functional role in the eye [30]. Finally, another protein exclusively present in stimulated saliva samples was proteasome subunit alpha type 5 (PSA5). It is a component of the 20S core proteasome complex involved in the proteolytic degradation of most intracellular proteins.

Differently, two proteins were shown to be upregulated while two others were downregulated in the stimulated saliva samples as compared with the unstimulated ones. Namely, the first two were glycosyl-phosphatidylinositol-anchored molecule-like protein (GML) and heat shock protein beta-3 (HSPB3). GML plays a crucial role in the mechanism of apoptosis or in the regulation of the cell cycle, following DNA damage. As well, HSPB3, present in different tissues, mainly muscle, is one of the small heat shock proteins (sHSPs) involved in the regulation of a number of cell functions such as cell cycle, apoptosis, differentiation, and signal transduction and in maintaining cytoskeletal integrity [31]. In contrast, the downregulated proteins were two isoforms of bactericidal proteins of the fold-keeping family (BPIF), which are abundantly expressed in human salivary glands. These proteins play a crucial role in the innate systemic immune response to Gram-negative bacteria. In particular, isoform 2 (BPIFA2) appears to be a unique protein for the oral cavity, where it produces bactericidal effects against Pseudomonas Aeruginosa. Thus, their downregulation in saliva samples of apparently healthy subjects is not surprising, even though they may be down-regulated in oral cavity diseases, such as in Syogren syndrome [32].

### 2.5. Bioinformatic Analysis of the Proteins Extracted from Unstimulated and Stimulated Saliva Samples, Selected from 2DE Electrophoretic Runs and Analyzed by MS

To better understand the possible relationship between the salivary proteins characterizing our samples, we imported the MS identification data into the STRING database. The analysis performed by this system allowed us to construct a network among all proteins identified by MS in both types of saliva samples and reported in Table 8. Some of the identified proteins show a more marked connection, as highlighted by the bold line linking these proteins (Figure 8A).

Furthermore, we asked the database to build the most relevant clusters among all the proteins listed in Table 8. With this analysis, the proteins were grouped in three clusters (Figure 8B). Cluster n. 1 included proteins coupled to the Rho GTPase pathway activating WASP (Wiskott–Aldrich syndrome protein) and WASP family verprolin-homologous proteins (WASP). This biochemical system is involved in the regulation of a wide variety of cell processes ranging from cell morphology to cell survival, proliferation, and adhesion [33]. In contrast, cluster n. 2 included proteins closely related to one of the saliva functions, specifically the sensory perception of bitter taste, while cluster n. 3 includes three proteins (see the table below Figure 8B) that may be related to immune-innate response [34].

STRING analysis also allowed us to determine the enrichment of tissue expression of genes that interact with those encoding proteins identified in saliva samples. This analysis confirmed that these proteins are interconnected with many others primarily present in the oral tract (including salivary glands, mouth, saliva, and even tears). Interestingly, most of the genes encoding the detected proteins are related to the glands, while some of them encoding proteins such as PSA5, PIGR, S100A9, ILEU (encoded by SERPINB1), ACTG1, and ACTB can interact with genes belonging to the respiratory system and, therefore, if altered, could be indicators of cellular alteration in this anatomical tract (Figure 9A). Enrichment related to biological processes confirmed that most of the identified proteins are related to bitter taste perception and some of them to transepithelial transport. Interestingly, some proteins might be involved together with others in retinal homeostasis (Figure 9B).

### 2.6. Comparative Analysis Between the Salivary Proteome of CT-Negative and CT-Positive Subjects for the Presence of a Solitary Pulmonary Nodule

Here, we report, only as an example, the results of some pilot experiments in which we performed a comparative analysis of only stimulated saliva samples from subjects negative (C, control) and positive for nodular lesions (N, nodular), as assessed by CT. Using stimulated saliva samples and the gel-based approach, 2DE maps were produced for each condition. A total of 25 µg of total proteins was resolved on a gradient of pH 4–7, using a 7 cm long gel strip (Figure 10A). The 2D run, on a 12% homogeneous gel, resolved 328 ± 71 and 304 ± 55 spots for the C and N samples, respectively.

Among these, we detected 44 unmatched spots in the salivary proteome of a smoker subject without nodular lesions, likely representing constitutive salivary proteins. We also identified 23 spots of unmatched proteins in the saliva samples of a patient with a nodular lesion, which should be examined more closely as they may be potential tumor biomarkers. Through subsequent MS analysis of the excised protein spots, we identified several interesting proteins, particularly the enzymes SERPB3 and aldoketoreductase family 1, isoform A1 (AKR1A1), as well as another protein from the Annexin family, isoform S100A14 (S100A14). Of these, SERPB3 was upregulated in stimulated saliva obtained from a patient with a nodular lesion. Of note, this protein acts as a papain-like cysteine protease inhibitor to modulate the host immune response against tumor cells (reviewed in [35]). Furthermore, among the other unmatched proteins, we detected AKR1A1. This enzyme catalyzes the NADPH-dependent reduction in a variety of aromatic and aliphatic aldehydes to their corresponding alcohols and plays a role in the activation of procarcinogens, such as polycyclic aromatic hydrocarbons trans-dihydrodiols, and in the metabolism of various xenobiotics and drugs. Therefore, AKR1A1 can have both protective and detrimental effects in cancer. While it can detoxify harmful substances and reduce oxidative stress, its ability to metabolize drugs and potentially activate carcinogens can contribute to cancer development and resistance to treatment [36]. We also revealed the selective presence of S100A14 in saliva from N patient. This protein is differentially expressed in various normal human tissues while its deregulated expression appears to be a common event in human carcinogenesis. Functionally, S100A14 has been linked to a variety of cellular activities related to carcinogenesis, such as cell proliferation and apoptosis, tumor cell migration and invasion, and keratinocyte differentiation. In particular, S100A14 has been reported to be overexpressed in lung adenocarcinomas compared to normal control tissues and appears to predict poorer survival [37]. The increased content of these three proteins in the stimulated saliva sample of a patient with a CT-positive nodule compared to the same proteins detected by specific antibodies in the saliva of a control subject was confirmed by Western blot analysis (Figure 10B).

## 3. Discussion

LC has a high mortality rate, often due to late diagnosis. Unfortunately, this tumor is currently discovered using invasive and potentially harmful techniques, such as CT. The recent adoption of LDCT has improved the visualization of suspicious nodules while reducing the toxicity, as well as new imaging software appears to improve the accuracy of standard CT scans [38,39]. Nonetheless, the need for new methods to accelerate the discovery of LC and improve patient outcomes is growing. Liquid biopsy, combined with multi-omics analysis of biological fluids collected from patients suspected of having cancer, may represent a new, noninvasive method for detecting potential biomarkers useful for early cancer screening, including LC [40]. Although the routine use of this integrated method needs to be standardized, we were intrigued by this framework and sought to develop a cost-effective, noninvasive technique that would allow for the identification of potentially useful biomarkers for early diagnosis of LC.

We primarily used a proteomic approach to analyze proteins extracted from saliva samples collected by passive drool or salivary swabs from healthy elderly patients. These two methods provide different types of saliva but the superiority of one over the other as well as the guidelines for their storage have not yet been well defined [41]. Therefore, in the first part of our study, we sought to determine the best procedures for the preservation of our sample and protein extraction from both of them. Our results indicated that salivary proteins can be altered during prolonged cryopreservation even at a temperature of −80 °C, which is considered suitable for the preservation of saliva samples [42]. Indeed, it has been reported that the saliva storage even for brief periods (7 to 30 days) alters the structures of the examined proteins due to their degradation. However, these events were observed when the samples were cryopreserved at 4 °C or −20 °C, and not at −80 °C. Differently, another study examining tissue samples rather than biological fluids, showed that protein concentration decreases after long periods of sample cryopreservation, at least at −80 °C, associated with an increase in microbial flora [43]. This may have contributed to altering the proteins in our unstimulated saliva samples, which are richer in bacteria, after varying periods of cryopreservation. However, even the use of Salivettes, which allows us to obtain samples richer in specific salivary proteins, helping to filter contaminants [40], reduced but did not eliminate possible contaminants, altering the protein stability during cryopreservation. Finally, also the adoption of a recently developed technology named “protein equalization”, specific for the enrichment of LAPs and increase in the specificity and sensitivity of proteomic analysis, was found to not be useful, being unable to improve the visualization of LAPs, masked by the predominance of HAPs, in the saliva. Therefore, we decided to directly analyze saliva samples from which proteins were extracted after a short cryopreservation period.

Based on this protocol, we used MS to analyze only proteins with the most significant content among those resolved by 2DE from both types of saliva samples. Many of them were typically produced by salivary glands or were related to the function of an oral apparatus and glands of the superior body tract. Bioinformatic analysis confirmed that some of them are implicated in bitter taste perception and transepithelial transport. These data are not surprising as some of the proteins we have identified are rather specific to saliva and derive from the upper body tract.

Interestingly, this analysis also highlighted that the proteins reported in Table 8 can be grouped into three clusters. While six of them, included in cluster n. 2, were confirmed to play a role in bitter taste perception, nine were grouped in cluster n.1 (Figure 8B) and belong to the Rho GTPases-WASP and WAVE pathway. This biochemical cascade plays a crucial role in cancer [44,45]. In particular, some proteins downstream from the Rho GTPase cascade are related to LC cell invasiveness (reviewed in [46]) and poor clinical outcome of patients with non-small-cell lung cancer (NSCLC) [47]. Finally, the last three proteins were included in cluster n. 3; two of them more properly belong to the Serpin family, which might be involved in preventing damage caused by pathogens [48], while the other, S100A9 protein has been shown to be implicated, when upregulated, in the progression of several tumors, including LC, in particular, NSCLC [49]. This protein, together with S100A8, can also be expressed in LC tissue and is related to differentiation [50].

In agreement with these findings, further literature data have indicated that a number of the proteins identified in our unstimulated or stimulated saliva samples have been found to be dysregulated in other biological fluids or tissues in patients with tumors, including LC. For example, ENOB is also considered a potential prognostic biomarker for tumors because it promotes cell growth, migration, and invasion. In particular, a recent review highlighted the expression of ENOB with other three proteins, namely BASP-1 (brain acid soluble protein 1), AL1A1 (aldehyde dehydrogenase 1A1), and SEGN (secretagogin), in both small-cell lung cancer (SCLC) and large cell neuroendocrine carcinoma (LCNE) [51]. Again, the gene CST1, encoding for cystastin SN, is correlated with poor patient outcome when upregulated in LC [27]. Alpha-amylase is also released into the serum and urine when a malignant neoplasm develops such as LC, ovarian cancer, or thymoma [52]. In particular, AMY1A is involved in oral cancer [53], and most lung tumors that produce amylase have been identified histologically as adencarcinomas [54]. Therefore, this amylase isoform could constitute a real proliferation biomarker, since it reflects the tendency of lung cells to differentiate towards a cell type much more similar to bronchial epithelium [55]. Differently, the neutrophil elastase ILEU, likely responsible for tissue injury in human inflammatory diseases such as respiratory disease [56] also shows an increased activity in LC patients [57]. Likewise, the presence of an isoform of Heat Shock Proteins (HSPs) appears to be essential not only in physiological processes (i.e., cell cycle regulation, differentiation, and apoptosis) but also in carcinogenesis, mainly related to the oral cavity [58], although they were also found to be dysregulated in LC models [59,60]. As for PSA5, also this subunit, when overexpressed, was found to be associated with LC [61]. Another interesting protein, ERI1, expressed at the nuclear level, has also been identified as a probable biomarker of lung carcinogenesis, especially in the phases of acquisition of aggressiveness of the tumor phenotype, regulating and controlling the angiogenesis mechanisms that predispose to the metastasis of LC [62]. As well, recent evidence suggests that elevated expression of SERPINB3 together with SERPINB4 is not limited to cancers of squamous origin but also extends to adenocarcinoma of the lung, breast, and pancreas, as well as hepatocellular carcinoma [35]. Furthermore, SERPINB3/B4 expression correlates with resistance to platinum combined treatment in NSCLC and poor prognosis of anthracycline-based chemotherapy in breast cancer patients [35].

Therefore, our results reported above, as well as literature data, which have previously shown the possibility of isolating potential candidate biomarkers for LC [13,63] from saliva, encourage us to proceed with our study on this biological fluid, extending the proteomic analysis to samples obtained from patients with suspicious lung nodules. In this way, it could be possible that the proteomic analysis of these saliva samples may allow some of the abovementioned proteins to be identified as dysregulated in patients with suspicion of LC. In this regard, the results reported as preliminary, which were obtained by the examination of the saliva sample from a patient with a lung nodular lesion (N), seems to also prompt us to continue. Indeed, and just for example, we detected the upregulation of SERPB3 as well as the exclusive presence of S100A14 and AKR1A1 proteins in the saliva of this patient, which have already been implicated in LC and could represent potential lung cancer biomarkers.

Obviously, we recognize that validating our method requires examining saliva samples from a large population of patients with different types of suspicious pulmonary nodular lesions. We have already initiated this second study, hoping to confirm the utility of proteomic analysis of salivary proteins as a new diagnostic test for the early diagnosis of LC.

## 4. Materials and Methods

### 4.1. Materials and Instruments

Salivette^®^ devices, purchased from Sarstedt (Numbrecht, Germany), were used for saliva collection, along with Falcon (50 mL) as sterile containers. Protease inhibitor mixture, DryStrip 4-7 IPG immobilins (acrylamide gel strips), Dry Strip cover fluid (98% liquid paraffin solution), Reydratation Solution (a running buffer for IEF), IPG buffer 4-7 (40% ampholin mixture), DeSTREAK reagent (hydroxyethyl disulfide), and agarose were from GE Healthcare (Uppsala, Sweden). 3-3(cholamidolpropyl)diethylammonium-1-propane sulfonate (CHAPS), tributyl phosphine (TBP), acrylamide, piperazine diacrylamide (PDA), sodium dodecyl sulfate (SDS), ammonium persulfate (APS), tetramethylenediamine (TEMED), glycerol, bromine phenol blue, tris(hydroxymethyl)aminomethane (TRIS), glycine, dithiothreitol (DTT), iodoacedamide (IAA), silver nitrate, formaldehyde, glutaraldehyde, ammonium bicarbonate, alpha-cyano-4-hydroxycinnamic acid (CHCA), dihydroxybenzoic acid (DHB), trypsin, and calibration peptides for mass spectrometry analysis were purchased from Sigma Chemical (St. Louis, MO, USA). 2-7-naphthalene disulfonic acid (NDS) was from Acros, Thermo Fisher Scientific (Geel, Belgium), and porcine trypsin was from Promega Bioscience, (San Louis Obispo, CA, USA). All reagents except those specified were supplied by Sigma Chemical (St. Louis, MO, USA). All solutions used were prepared with Milly-Q water (Millipore, Bedford, MA, USA).

As for the technical equipment, the first-dimensional run instrument, IPGphor III, was from GE Healthcare, Uppsala, Sweden, while the second-dimensional chamber was supplied by Biorad Laboratories (Hercules, CA, USA). After silver staining, the gels were scanned using LabScan (GE Healthcare, Uppsala, Sweden) and subsequently analyzed with the 2D analysis program Platinum 6.0 (GE Healthcare, Uppsala, Sweden). Mass spectrometry was performed using AUTOFLEX Speed MALDI LIFT-TOF/TOF MS (Bruker Daltonics, Bremen, Germany).

### 4.2. Saliva Collection and Sample Preparation

In this study, we analyzed saliva samples collected from 182 healthy subjects enrolled for a screening among a population of long-time heavy smokers but without signs of lung cancer. For this preliminary study of a methodological set up, the subjects were females/males aged 72.98 ± 9.35 years who provided written informed consent before saliva collection. They reported no chronic illnesses or psychiatric disorders requiring medication, and no oral conditions (e.g., periodontal diseases and tooth pain) during the sampling period.

All participants were asked to abstain from brushing their teeth, eating, and drinking for at least 1 h before saliva collection. Participants rinsed their mouths with water and were instructed on how to collect saliva. The first method was drooling, which is commonly used to collect unstimulated saliva (also called “whole saliva” or “mixed saliva”) and requires subjects to salivate through a straw into a sterile vial [64]. In the second method, cotton-based collection devices (Salivette^®^) were used [65], which are generally used to obtain (mechanically) stimulated saliva through chewing a swab for 2 min. They provided 5 mL of unstimulated whole saliva via passive drooling over a period of 15 min and, subsequently, 1–2 mL of stimulated saliva samples using the Salivette^®^ device.

The collected samples were placed on ice during transport from the collection site to the laboratory. A total of 10 μL of sodium azide (NaN_3_) and 40 μL of protease inhibitors (Complete^TM^ Protease Inhibitor Cocktail, Cat.1697498, Roche, Mannheim, Germany) were added to each unstimulated saliva sample that was centrifuged at 1200 rpm for 5 min to remove interferents; similarly, 2 μL of NaN_3_ (0.01 mM) and 10 μL (100X) of protease inhibitors (Protease Inhibitor Mix, GE Healthcare, Düsseldorf, Germany) were added to each sample contained in the Salivette that, after mixing and centrifugation at 4000 rpm at 5 °C for 5 min, was recovered from the swab in a total volume equal to 1–2 mL, free from any interferents. Each sample, aliquoted to a final volume of 1 ml, was centrifuged at 12,800 rpm × 10 min at 4 °C. Once the pellet was separated, the supernatant thus obtained was stored at −80 °C until further analysis.

### 4.3. Protein Extraction from Stimulated and Unstimulated Saliva and Measurement of Their Amount

As previously reported [66], 500 μL of saliva from each sample were mixed with 1.5 mL of acetone and incubated overnight at −20 °C. Subsequently, samples were centrifuged for 15 min at 14,800 rpm; then, the supernatant was discarded, and the resulting pellet was resuspended in 445 μL of lysis buffer (R2: 5M Urea, 2M Thiourea, 40 mM Trizma base, 2% CHAPS and 2% SB3-10) (Sigma Aldrich, St. Louis, MO, USA) and mixed with 5 μL of ampholine biolyte 3-10, 5 μL of Pheny-methyl-sulfonifluoride (PMSF), 5 μL of tributylphosphine (TBP), and 5 μL of protease inhibitor cocktail. After vortexing for at least 1 h, each sample was subjected to 3 sonication cycles of 20 sec, followed by incubation at room temperature in 500U of Benzonase. After centrifuging each sample at 14,800 rpm for 15 min and eliminating the supernatant, each pellet was resuspended in 1.5 mL of acetone and was left to precipitate overnight at −20 °C. Finally, after another round of centrifugation for 15 min at 14,800 rpm, eliminating the supernatant, the pellet, once dried, was resuspended in R2 and TBP. The protein determination of each sample was carried out by the Better Bradford method.

### 4.4. 2DE Analysis

From each of the protein samples, obtained as described above, different amounts of proteins were used depending on the size of the gels used. For the 7 cm strip gels, 10 and 25 µg of proteins extracted from unstimulated and stimulated saliva samples, respectively, were loaded, while for the 24 cm gel (analytical gels), 150 and 80 µg of proteins extracted from unstimulated and stimulated saliva samples, respectively, were used according to the procedures previously described [67]. Typically, 3 independent runs were performed for each sample. A reference gel was constructed for each type of saliva, i.e., unstimulated and stimulated samples, by matching the 2D maps of the 3 individual electrophoretic runs, while a master gel was then prepared by matching all the reference gels for both unstimulated and stimulated saliva samples (Figure 6). A comparative proteomic analysis was performed on the latter using Image Master 2D Platinum software (version 6.0) to determine quantitative changes in protein expression. To this end, the intensity volume of each spot, after background subtraction, was normalized to the total intensity volume. The same software also allowed us to perform all statistical analyses including the FDR method to improve the reliability of the 2DE results, ensuring the validity of the protein identification.

### 4.5. Protein Digestion and MALDI-TOF/TOF-MS Analysis

All protein spots chosen according to their intensity levels, which were significantly different among the investigated groups, were excised from 2-D gels and analyzed by the peptide mass finger printing (PMF) approach using a MALDI-TOF/TOF spectrometer. Protein spots picked from the gel were washed with 100% ethanol and 100 mM ammonium bicarbonate (NH_4_HCO_3_) and then incubated at 56 °C in 100 µL of 50 mM NH_4_HCO_3_ containing also 10 mM dithiothreitol (DTT, 60 min) and subsequently at room temperature (30 min in the dark) in 100 µL of 50 mM NH_4_HCO_3_ plus iodoacetamide. The gel was then incubated in 50 mM NH_4_HCO_3_ containing trypsin at 37 °C overnight [68]. These peptide extracts were applied to a C18ZipTip (Millipore, Bedford, MA, USA), rinsed (0.1% trifluoroacetic acid, TFA), and eluted directly on the MALDI target in 0.5 μL of a saturated solution containing 1:1 α-cyano-4-hydroxycinnamic acid: 0.1% TFA. MS analysis was performed as previously described [69]. For high-precision calibration (HPC), a mixture of reference peptide fragments (i.e., bradykinin fragment 1–7, 757,39 *m*/*z*; angiotensin II 1046,54 *m*/*z*; ACTH fragment 18–39, 2465,19 *m*/*z*; GluFibrinopeptide B 1571,57 *m*/*z*, and renin substrate tetradecapeptide porcine 1760,02 *m*/*z*) was used, while for internal mass calibration, trypsin autodigestion products (843,50 *m*/*z*, 1.046,56 *m*/*z*, 2.212,11 *m*/*z*, 2.284,19 *m*/*z*) were used.

### 4.6. Database MS/MS Searching

Spectra obtained by PMF were searched in the NCBIn protein database by the Mascot search engine. Multiple parameters (i.e., PMF enzyme, trypsin; fixed modification, carbamidomethylation (Cys); variable modifications, oxidation of methionine; mass values, monoisotopic; ion charge state was set to +1, maximum mis-cleavage was set to 1; mass tolerance of 100 ppm for PMF and 0.6–0.8 Da for MS/MS) were used in this search to compare the experimentally determined tryptic peptide masses with theoretical peptide masses calculated for proteins from the database. Additionally, after automated assessment of the search results, the samples were also submitted to LIFT TOF/TOF acquisition for validation of data analysis from PMF, as previously described [70]. A maximum of four precursor ions per sample were chosen for MS/MS analysis. A further protein database search using combined PMF and MS/MS datasets was performed using the SwissProt database (SwissProt_2012_03.fasta) via BioTools 3.2 (Bruker Daltonik GmbH, Bremen, Germany) connected to the Mascot search engine. The Mowse probability score (the observed match between the experimental dataset and each sequence database entry is a chance event, *p* < 0.05) was used as a criterion for correct identification. Scores are reported as −10 Log_10_(p), where p is the probability. The lowest probability corresponds to the highest score and is reported as the best match. Typically, this is a score of about 70 for PMF and 30–40 for MS/MS search.

### 4.7. Bioinformatics Analysis of Proteomic Data

Functional protein association was performed using the STRING software (http://string-db.org, version 11.0). This program builds protein networks based on known direct and indirect interactions described in the literature. A confidence level of 95% was set as a cut-off for the bioinformatics analysis.

### 4.8. Western Blot Analysis

Stimulated saliva samples obtained from a healthy subject and a patient CT-positive for a lung nodule were used. Protein samples (usually 50 μg), diluted in sodium dodecyl sulphate (SDS)-bromophenol blue buffer and boiled for 5 min, were run on 12% SDS polyacrylamide gels and subsequently transferred on polyvinylidene fluoride (PVF) membranes. After protein blockade by a mixture containing PBS, 0.1%, Tween20, and 5% nonfat milk (Bio-Rad Laboratories, Hercules, CA, USA) for 2 h at 4 °C, PVF membranes were incubated (overnight, 4 °C) with primary polyclonal antibodies: rabbit anti-S100A14 (PA5-78543, diluted at 1:500) purchased from ThermoFisher Scientific; rabbit anti-SERPINB3 (PA5-30164, diluted at 1:1000) purchased from Invitrogen; rabbit Anti-AKR1A1 (ab251727, diluted 1:1300) purchased from Abcam, and subsequently (1 h, room temperature) with goat anti-rabbit HPR-conjugated secondary antibody (final dilution 1:5000, Bethyl Laboratories Inc.; Montgomery, TX, USA). Immunocomplexes were visualized by chemiluminescence (ECL) detection system (GE Healthcare Life Sciences, Milan, Italy).

### 4.9. Statistical Analysis

The reference gels as well as synthetic gels were used to evaluate the presence of and difference in protein levels. Background subtraction was performed, and the intensity volume of each spot was normalized by total intensity volume (summing the intensity volumes obtained from all spots within the same 2-D gel). All the quantitative data are reported as mean ± SD values. The intensity volumes of individual spots were matched across the different gels and then compared among groups by multiple comparisons using one-way analysis of variance (ANOVA). A *p* value < 0.005 was considered statistically significant. Significantly different protein spots were subjected to in-gel tryptic digestion and identification by MS. Furthermore, statistical comparisons between values from different treatments in the same model were calculated using Student’s *t*-test for unpaired data by GraphPad Prism software (version 6.0). *p*-values were corrected for multiple comparisons when appropriate.

## 5. Conclusions

In this article, we explored the potentiality of the saliva proteome analysis to be used as more sensitive and less expensive tests than CT to identify lung tumors in preclinical phase.

Our work is part of a regional project aimed at identifying and validating tumor-associated molecular markers in the saliva of heavy smokers enrolled in the screening program. To this end, experiments are underway on saliva samples obtained from a large number of patients suspected of having LC. Patients will be or have already been enrolled based on routine CT scans and physical examinations performed by lung oncology specialists. Based on their diagnosis, patients have been or will be divided into healthy individuals and those with suspicious tumor nodules, who will in turn be divided based on the nature of the nodule(s). Consequently, all data obtained from the analysis of their saliva samples will be compared in the hope of identifying potential tumor biomarkers and subjected to appropriate statistical analysis. Since we anticipate a large number of patients, we will decide whether it is possible to also stratify the population based on age, dental health, overall duration of smoking, and the presence of other medical conditions. Patients with previous tumor lesions outside the lung will be excluded from this study.

For the first time in our region, the effectiveness of a LC screening program is being tested using a proteomic approach. If our method is validated, all the results obtained could be entered into a regional oncology database, which we hope will provide a useful platform for integrating genomic and proteomic data with clinicopathological and molecular data from patients with precancerous lesions and/or confirmed malignancies.

We hope this study will attract the attention of other research groups and that our continued studies will lead to improved diagnostic techniques for LC, making them less invasive and more targeted, thus increasing the chances of timely treatment and improving patients’ quality of life.

## Figures and Tables

**Figure 1 ijms-26-07924-f001:**
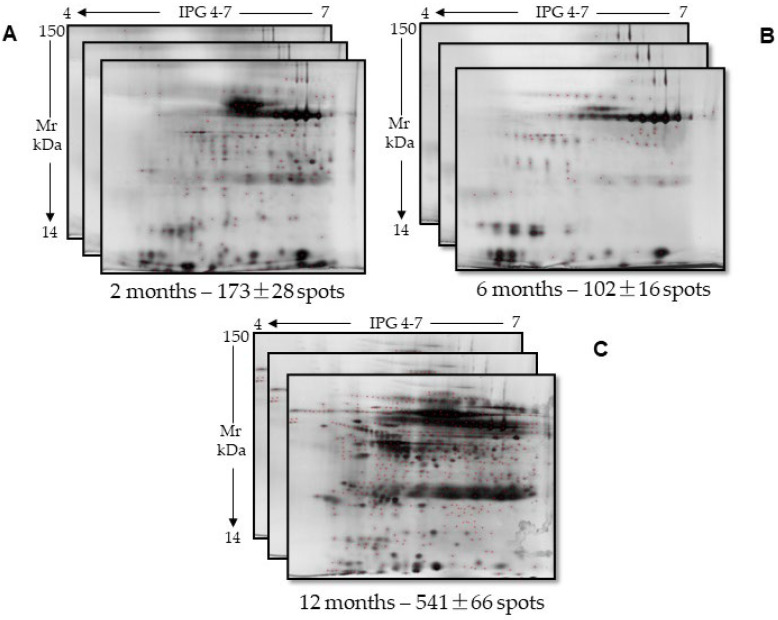
Two-dimensional maps of technical triplicates of 3 different cryopreserved samples: (**A**) 83C, at 2 months; (**B**) 51C, at 6 months; and (**C**) 11C, at 12 months. The mean number of total identified spots ± standard deviation is reported below each set of 2D maps.

**Figure 2 ijms-26-07924-f002:**
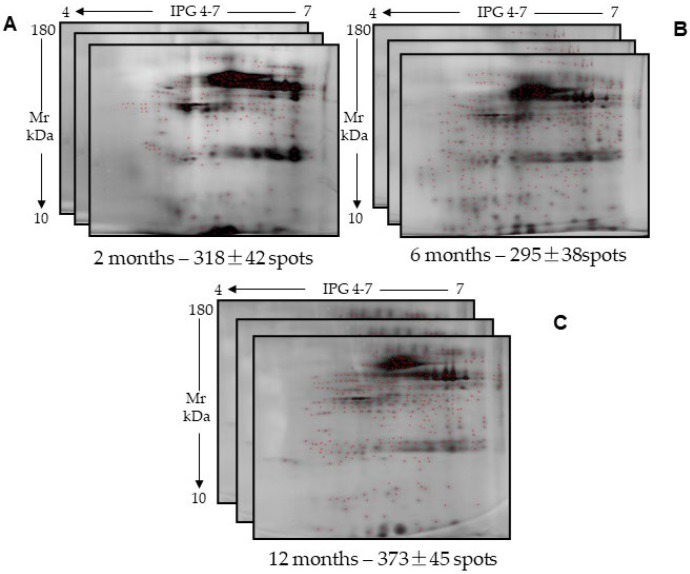
Two-dimensional maps of technical triplicates of the sample 11C, which was extracted and cryopreserved for (**A**) 2 months; (**B**) 6 months; and (**C**) 12 months. Below each panel, the duration of cryopreservation time and the number of identified spots ± standard deviation are reported. There was no statistically significant difference among the number of spots identified within each of the three technical replicates, as evaluated by Student’s *t* test.

**Figure 3 ijms-26-07924-f003:**
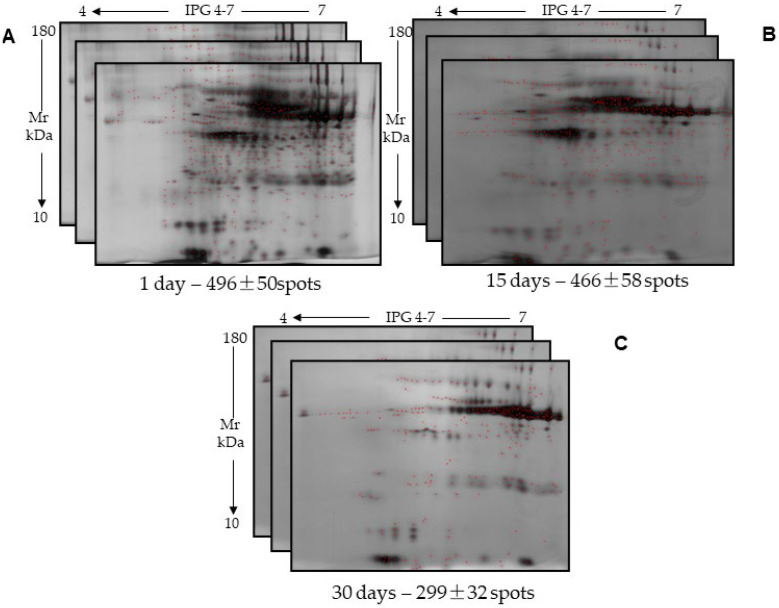
Two-dimensional maps of technical triplicates of saliva samples collected by Salivette, which were cryopreserved for (**A**) 1 day; (**B**) 15 days; and (**C**) 30 days and from which the proteins were extracted at the end of the cryopreservation period. Below each panel, the duration of cryopreservation time and the number of identified spots ± standard deviation are reported. There was no statistically significant difference among the number of spots identified within each of the three technical replicates, as evaluated by Student’s *t* test.

**Figure 4 ijms-26-07924-f004:**
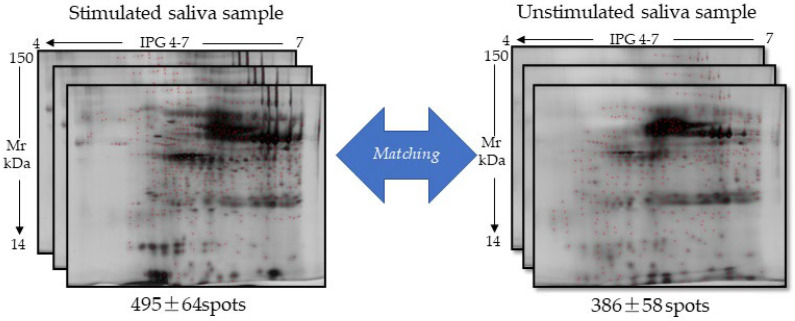
Comparative analysis in technical triplicate between saliva collected with Salivette (stimulated sample) vs. unstimulated saliva sample. Below each panel, the duration of cryopreservation time and the number of identified spots ± standard deviation are reported. There was no statistically significant difference among the number of spots identified within each of the three technical replicates, as evaluated by Student’s *t* test.

**Figure 5 ijms-26-07924-f005:**
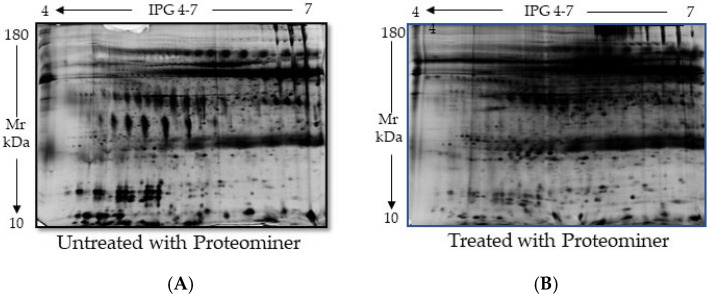
Two-dimensional analysis of the proteome obtained from stimulated saliva samples submitted to a protein enrichment process (Proteominer). (**A**) Saliva not treated with Proteominer. (**B**) Saliva treated with Proteominer. A total of 150 µg of saliva was resolved on 24 cm gel strips (pH 4–7) and gradient gels (9–16%).

**Figure 6 ijms-26-07924-f006:**
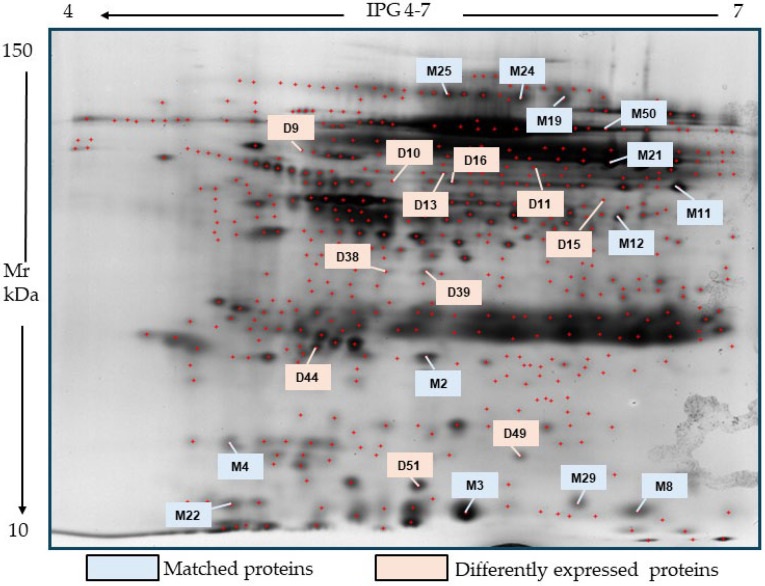
Reference 2D map obtained by comparing 2D maps obtained from electrophoresis of 170 unstimulated and stimulated saliva samples. Numerous proteins distributed on homogeneous gels (12%) at pH 4–7 were resolved, and then, a great number of them were identified by MALDI TOF/TOF MS analysis. Of these, proteins with the highest expression were labeled as matched (M), present in equivalent amounts in unstimulated and stimulated saliva samples, or differently expressed (D). A list of these proteins is reported in Table 8 (see below).

**Figure 7 ijms-26-07924-f007:**
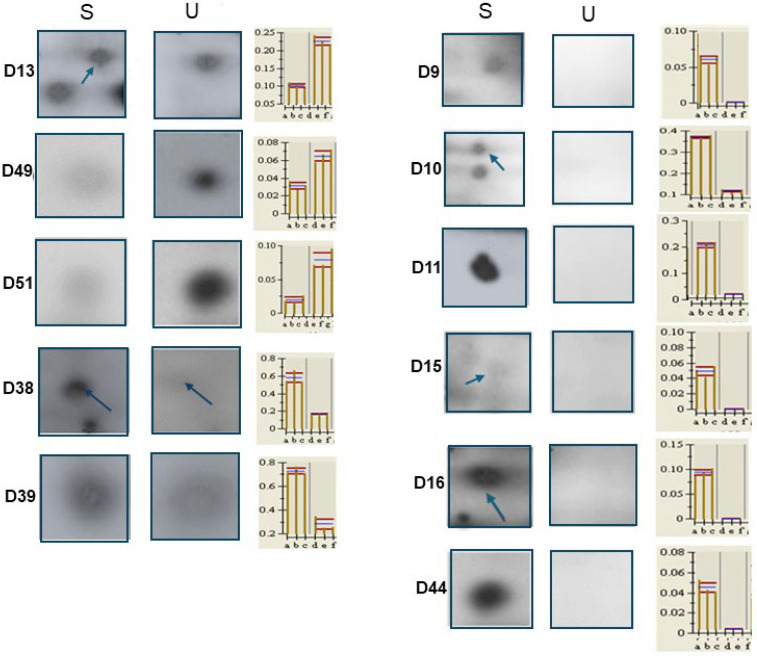
Magnification of spots obtained from 2DE gels. This technique resolved proteins extracted from stimulated (S) and unstimulated (U) saliva samples. Only proteins showing different expressions in the two sample types are shown here, while the histograms with the corresponding spot intensity volume values are shown on the right of the spot panels. Arrows indicate the selected spots when there are more than one or they are poorly visible.

**Figure 8 ijms-26-07924-f008:**
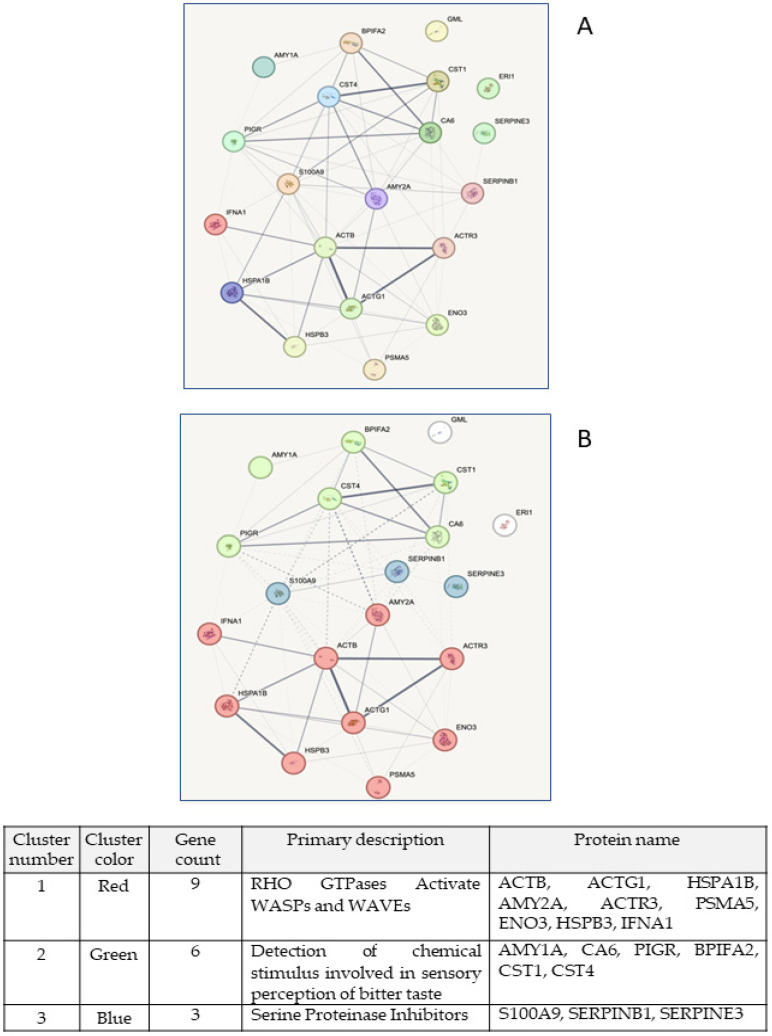
Computer analysis performed using the STRING database to evaluate the mutual interactions between the proteins identified by MS analysis and reported in Table 8. (**A**) This analysis highlighted a close interaction between the genes encoding the most significant proteins identified by MS in the saliva of healthy subjects. (**B**) STRING analysis divided these genes and related proteins into three clusters, where the gene names are reported close to circles labelled with different colors and whose composition is reported in the Table below the graph. The names of the genes and proteins encoded by these genes are reported in Appendix A.

**Figure 9 ijms-26-07924-f009:**
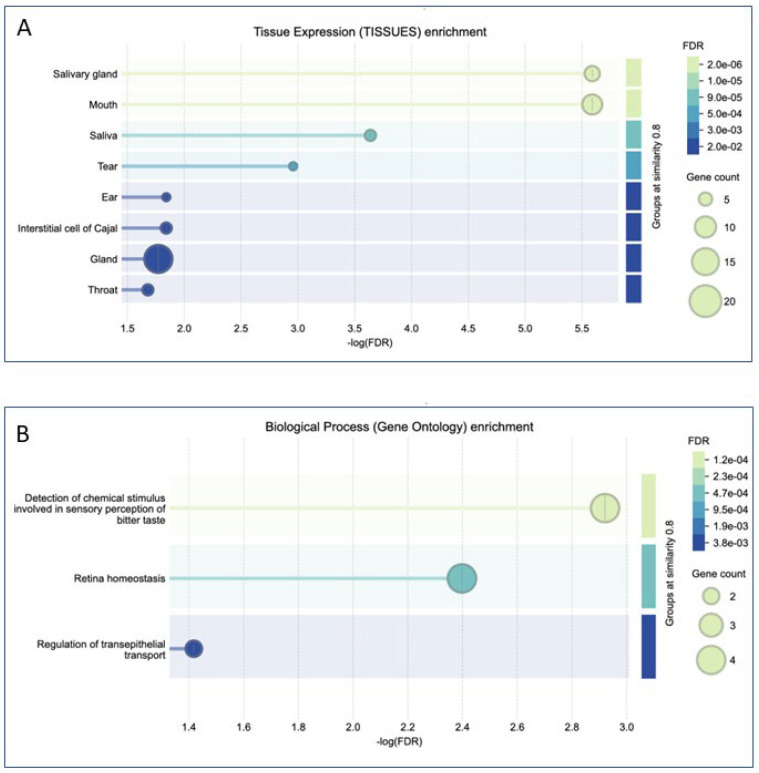
Enrichment analysis of tissue expression (**A**) and biological processes (**B**) in which all proteins included in Table 8 are present or involved in, respectively. The abscissa reports −log FDR (False Discovery Rate), which represents the expected proportion of false positives among the gene sets declared as significant. On the right of each panel, the colored bar ranging from blue to green represents the FDR, expressed as –log10, showing decreases to increases for the proteins in the groups, while the green circle size represents the number of genes involved in this analysis.

**Figure 10 ijms-26-07924-f010:**
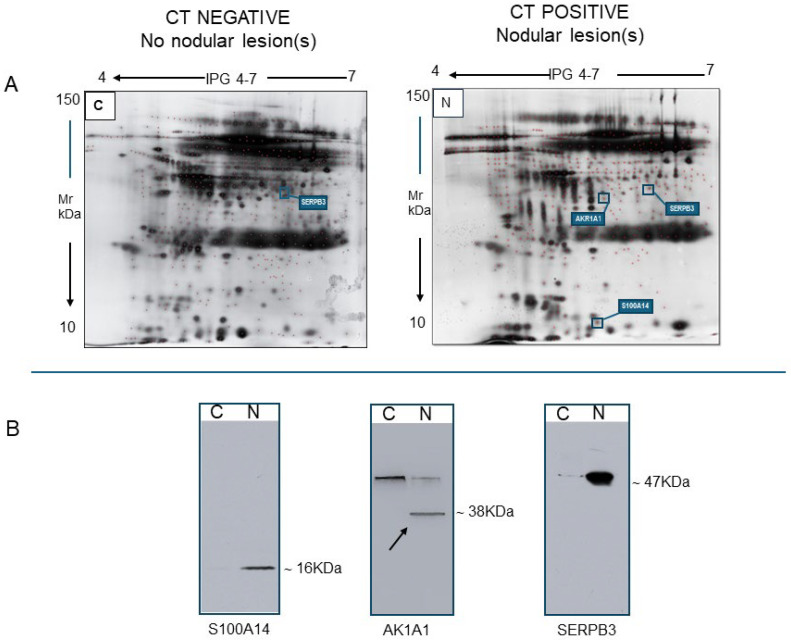
Comparative analysis between proteins extracted from stimulated saliva samples collected from a control subject, negative for lung nodular lesions, and a patient with lung nodular lesion at CT. (**A**) Representative 2D maps of proteins from these two saliva samples. A total of 25 µg of saliva was applied on 7 cm long strips at pH gradient 4–7. (**B**) Representative Western blots obtained from proteins extracted from the two saliva samples and loaded into each electrophoresis lane (30 µg). They show variations in the protein content of serpin B3 (SERPB3), aldoketoreductase family 1, isoform A1 (AKR1A1) (the white arrow indicates the appropriate immune band), and S100A14 in the two samples. The approximate molecular weight of the identified proteins is reported next to the corresponding bands. C: control. N: nodular lesions.

**Table 1 ijms-26-07924-t001:** Mean unit concentration of proteins extracted from unstimulated salivary samples after 2, 6, and 12 months of cryopreservation.

Sample	Duration of Cryopreservation	Mean (µg/µL)	SD
83C 45C 77C	0 months (T_0_)	3.124	0.23
	2 months	3.023 (NS)	0.11
51C 44C 68C	T_0_	2.968	0.15
	6 months	2.054 (***)	0.14
11C 17C 27C	T_0_	3.57	0.12
	12 months	2.54 (***)	0.16

Samples were randomly chosen among those identified by a number assigned during their collection. Protein assay was performed in triplicate on 500 µL for each sample of unstimulated saliva while the mean protein concentration of the samples indicated for each cryopreservation period is reported in the third column. C: control; SD: standard deviation. Statistical significance: NS: not significant; *** *p* < 0.01 vs. the respective value at T_0_ (Student’s *t* test).

**Table 2 ijms-26-07924-t002:** Comparative analysis between 2DE maps of proteins extracted from the salivary fluids at the end of different cryopreservation times (2, 6, and 12 months).

Sample Used for 2D Gels	Cryopreservation Period	All Spots	# Matched Spots (vs. 12-Month Sample)	% Matched Spots (vs. 12-Month Sample)
83C	2 months	173	92	17
51C	6 months	102	67	12
11C	12 months	541	541	100

**Table 3 ijms-26-07924-t003:** Mean unit concentration of proteins extracted from an unstimulated salivary sample before cryopreservation for 2, 6, and 12 months.

Sample	Duration of Cryopreservation	Mean (µg/µL)	SD
11C	0 months (T_0_)	3.640	0.11
11C	2 months	3.174 (*)	0.17
11C	6 months	2.721 (**)	0.04
11C	12 months	2.608 (***)	0.20

The selected sample was identified by a number assigned during the collection as 11C, and the protein assay was performed in triplicate on 500 µL of the same sample of unstimulated saliva before cryopreservation. C: control; SD: standard deviation. Statistical significance: * *p* < 0.05; ** *p* < 0.002; *** *p* < 0.001 (Student’s *t* test) vs. the respective value at T_0_.

**Table 4 ijms-26-07924-t004:** Comparative analysis between 2DE maps of proteins extracted from the same salivary fluids before their cryopreservation for different periods.

Sample Used for 2D Gel	Cryopreservation Period	All Spots	# Matched Spots (vs. 12-Month Sample)	% Matched Spots (vs. 12-Month Sample)
11C	(2 months)	318	150	40
11C	(6 months)	295	210	56
11C	(12 months)	373	373	100

**Table 5 ijms-26-07924-t005:** Mean unit concentration of proteins extracted from stimulated saliva samples at the end of each cryopreservation period.

Sample	Duration of Cryopreservation	Mean (µg/µL)	SD
C31S C44S C53S	1 day	1.940	0.097
C86S C89S C101S C95S	15 days	1.750 (**)	0.11
C128S C132S C141S	30 days	1.563 (***)	0.080

Samples were identified by the number assigned during the collection by Salivette, a method allowing the saliva to be filtered during collection. Protein assay was performed in triplicate on 500 µL of the same samples of stimulated saliva. C: control. S: filtration through Salivette. SD: standard deviation. Statistical significance: ** *p* < 0.01; *** *p* < 0.001 (Student’s *t* test) vs. the mean value at 1 day.

**Table 6 ijms-26-07924-t006:** Comparative analysis between salivary fluids obtained by Salivette collection and from which proteins were extracted at the end of different cryopreservation times (1, 15, and 30 days).

Sample Used for 2D Gel	Cryopreservation Period	All Spots	# Matched Spots (vs. 1 Day Sample)	% Matched Spots (vs. 1 Day Sample)
C31S	1 day	496	496	100
C86S	15 days	466	255	51
C128S	30 days	299	170	34

**Table 7 ijms-26-07924-t007:** Comparative analysis of the spot number identified in 2DE maps obtained from the electrophoretic run of proteins extracted from unstimulated and stimulated saliva samples.

Sample Used for 2D Gel	All Spots	# Unmatched Spots (vs. 153S Sample)	# Matched Spots (vs. 153S Sample)	% Matched Spots (vs. 153S Sample)
125C	386	126	260	52.5
133C	307	180	127	25.6
137C	295	82	213	43.0
C147S	468	102	366	73.9
C149S	397	59	338	68.3
C153S	495	0	495	100

**Table 8 ijms-26-07924-t008:** Matched (M) and differently expressed (D) proteins extracted and identified from unstimulated and stimulated salivary fluid samples.

Label	Protein Name	Abbreviation	Swiss-Prot/NCBInr Ac ^a^	Score ^b^ _Sc ^c^	Matching Peptides #	Theoretical Mr_pI	Expression in Stimulated (SS) vs. Unstimulated Samples
M2	Interferon alpha-1/13	IFNA1	P01562	45_32	5	22,110_5.32	_
M3	Protein S100 A9	S100A9	P06702	101_90	13	13,291_5.71	_
M8	Cystatin SN	CYTN	P01037	70_68	11	16,605_6.73	_
M11	Alpha-enolase isoform 3	ENOB	P13929	102_67	27	47,696_6.5	_
M12	Carbonic anhydrase 6	CAH6	P23280	94_25	11	35,459_6.51	_
M19	Polymeric immunoglobulin receptor	PIGR	P01833	93_39	29	84,429_5.58	_
M21	Alpha-amylase 1A	AMY1A	P0DUB6	160_49	27	58,415_6.47	_
M25	Polymeric immunoglobulin receptor	PIGR	P01833	40_43	21	84,429_5.58	_
M22	Cystatin S	CYTS	P01036	83_73	13	16,489_4.95	_
M24	Heat shock protein 70 kDa protein 1A	HS71A	P0DMV8	126_39	19	70,294_5.48	_
M29	Cystatin SN	CYTN	P01037	138_37	4	16,605_6.73	_
M50	Pancreatic alpha-amylase	AMYP	P04746	160_49	27	58,415_6.47	_

D9	Leukocyte elastase inhibitor	ILEU	P30740	54_39	19	42,829_5.90	Only in SS (*p* < 0.0037)
D10	Actin, cytoplasmic 1	ACTB	P60709	100_48	27	42,052_5.29	Only in SS (*p* < 0.0006)
D11	3′5′ exoribonuclease 1	ERI1	Q81V48	37_29	17	40,494_6.29	Only in SS (*p* < 0.0018)
D13	Actin, cytoplasmic 2	ACTG	P63261	102_60	29	42,108_5.31	>expression in S (*p* < 0.0008)
D15	Serpin E3	SERP3	A8MV23	54_48	26	44,594_6.35	Only in SS (*p* < 0.0019)
D16	Actin-related protein 3	ARP3	P61158	85_53	35	47,797_5.61	Only in SS (*p* < 0.0024)
D38	BPI fold-containing family A member 2	BPIA2	Q96DR5	90_56	14	27,166_5.35	<expression in SS (*p* < 0.0005)
D39	BPI fold-containing family A member 2	BPIA2	Q96DR5	96_55	14	27,166_5.35	<expression (*p* < 0.0001)
D44	Proteasome subunit alpha type 5	PSA5	P28066	52_58	15	26,565_4.74	Only in SS (*p* < 0.0041)
D49	Glycosyl-phosphatidylinositol-anchored molecule	GML	Q99445	41_62	9	18,345_6.10	>expression in SS (*p* < 0.0007)
D51	Heat shock protein beta-3	HSPB3	Q12988	29_32	4	17,069_5.36	>expression in SS (*p* < 0.0029)

^a^: abbreviation of the protein name in the Swisse-Prot and NCBI databases; ^b^: score indicates the −10 × log P, where P represents the probability that the observed matching is a random event. It is calculated in the NCBI database, using the Mascot browser as a search engine; ^c^: Sequence Coverage indicates the ratio between the sequence segment covered by the common peptides, on the protein sequence as a whole. SS: stimulated saliva sample. *p*-values in parentheses represent the statistical significance of the difference in the amount of a protein identified by MS in stimulated versus unstimulated saliva samples, while the symbol “—“ indicates a lack of difference in the amount of protein between unstimulated and stimulated saliva samples.

## Data Availability

All data have been included in this article.

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
