# Peer review of "Methodological Development of a Test for Salivary Proteome Analysis Useful in Lung Cancer Screening"

_ijms, 2025, doi:10.3390/ijms26167924_

Round 1

Reviewer 1 Report

Comments and Suggestions for Authors

The authors present a study aimed at developing a proteomic methodology for identifying salivary biomarkers in lung cancer screening, particularly among heavy smokers. The topic is highly relevant and holds translational potential for non-invasive diagnostics. However, the manuscript requires substantial revision to clarify experimental design details, improve data presentation, address biological interpretation, and more rigorously validate findings.

While the proteomic workflow is technically sound, the novelty and translational significance of the results need better articulation. The identification of differentially expressed proteins is potentially valuable, but the biological and clinical relevance of several key findings remains speculative without adequate validation or contextual support.

Therefore, I recommend major revision before the manuscript can be considered for publication.

Major Issues to Address

1. Lack of Validation of Biomarkers

    • The study identifies numerous differentially expressed proteins in saliva, yet no orthogonal validation (e.g., western blotting or ELISA) is provided to confirm differential expression.

    • Validation, even on a subset of top candidate proteins (e.g., SERPINB3, AKR1A1, S100A14), is essential to strengthen the biological conclusions.

2. Insufficient Statistical Rigor

    • The manuscript lacks details on how the false discovery rate (FDR) was controlled during proteomic analysis.

    • Please provide volcano plots or other statistical visuals summarizing differential protein expression, along with adjusted p-values.

3. STRING and GO Analysis Need Clarification

  • The STRING network clustering is shown but not fully integrated with the discussion of biological significance.

  • Expand on the interpretation of enrichment results (e.g., bitter taste receptors, Rho GTPase signaling) and explain how they relate to lung cancer biology or oral immune responses.

4. Figures Need Improvement

    • The resolution and clarity of the 2D gel images (Figure 9) need enhancement. Annotate protein spots of interest for readers unfamiliar with this technique.

5. English Language and Flow

    • Several sections require careful language editing to improve clarity and reduce redundancy (e.g., long paragraphs with repetitive phrasing in the Discussion and Results).

    • Grammatical issues and awkward expressions occur throughout the manuscript (e.g., "mucosal immunity in shock protein" → unclear). A professional language edit is recommended.

Minor Suggestions

  • Reorganize the Discussion to avoid redundancy. Currently, the same proteins are repeatedly mentioned with overlapping functional speculation.

  • Specify how many total proteins were identified before filtering.

  • Clarify whether the STRING analysis included only significantly altered proteins or the entire saliva proteome dataset.

The manuscript presents promising preliminary data for salivary biomarkers in lung cancer screening. However, to elevate its scientific rigor and translational value, the authors must address the above major concerns—particularly validation of results, clinical relevance of sample groups, and improved presentation of bioinformatics and proteomics data.

Author Response

We thank this Reviewer very much for his/her valuable suggestions and constructive criticism. Below you will find our answers to the questions raised.

Major Issues to Address

  1. Lack of Validation of Biomarkers
  • The study identifies numerous differentially expressed proteins in saliva, yet no orthogonal validation (e.g., western blotting or ELISA) is provided to confirm differential expression.
    • Validation, even on a subset of top candidate proteins (e.g., SERPINB3, AKR1A1, S100A14), is essential to strengthen the biological conclusions.

R.: It was impossible for us to validate all the identified proteins reported in Table 8 of the manuscript by Western blot or ELISA tests, both due to the cost of the antibodies and all the materials required to perform these tests, and due to the limited time available, as this is a summer holiday period and the companies supplying such materials are closed or have a long time before they can deliver their products. Therefore, we validated only the three proteins identified as increased in the saliva sample from the patient with a nodular lung lesion by Western blot. The proteins in Table 8 were validated by LIFT technology, which allows us to confirm their protein sequence based on the high scores obtained by this test, reported in Table S1 of the Supplementary Material. We hope this will be sufficient.

  1. Insufficient Statistical Rigor
  • The manuscript lacks details on how the false discovery rate (FDR) was controlled during proteomic analysis.

R.: The FDR was calculated directly using Image Master 2D Platinum software (version 6.0), cited in the Methods section for 2DE analysis. 2DE is a powerful proteomics technique for separating proteins based on their isoelectric point and molecular weight from complex protein mixtures, enabling the identification of proteins differentially expressed in different conditions (healthy vs. pathological states). However, we recognize that mandatory statistical procedures, including normalization and multiple test corrections, are required during 2DE. FDR methods were applied to control the false-positive rate during these analyses, ensuring that the identified proteins are statistically significant. Furthermore, normalization procedures and statistical tests were evaluated to minimize the FDR. For example, using the moderated t-test in combination with the classic FDR method (Benjamini and Hochberg) has been shown to improve the reliability of results in 2DE analyses. A sentence has been added for this aspect in the Methods section (p. 23, lines 820-822).

  • Please provide volcano plots or other statistical visuals summarizing differential protein expression, along with adjusted p-values

R.: We have provided the magnification of the spots related to the proteins resolved by 2DE and then identified by MS analysis showing a different amount in stimulated saliva samples compared to unstimulated ones (see Figure 7). Furthermore, we have added a scatter plot analysis to demonstrate the validity of the identification of shared proteins in unstimulated and stimulated saliva samples confirmed by the correlation coefficient close to 1 (0.70) (see Figure S1 in the supplementary material and the explanation of the significance of this analysis in both the figure S1 caption and the manuscript text, page 12. We have also added the statistical significance of the increased content or exclusive presence of some proteins in stimulated saliva samples compared to those found in unstimulated saliva in the last column of Table 8.

  1. STRING and GO Analysis Need Clarification
  • The STRING network clustering is shown but not fully integrated with the discussion of biological significance.
  • Expand on the interpretation of enrichment results (e.g., bitter taste receptors, Rho GTPase signaling) and explain how they relate to lung cancer biology or oral immune responses.

R.: we have added sentences to explain the possible implication of some proteins grouped in different clusters in lung cancer (see text on page 20, lines 670-685)

  1. Figures Need Improvement
  • The resolution and clarity of the 2D gel images (Figure 9) need enhancement. Annotate protein spots of interest for readers unfamiliar with this technique.

R.: we reworked this figure, which is now n. 10, and we hope that it is clearer, now.

  1. English Language and Flow
  • Several sections require careful language editing to improve clarity and reduce redundancy (e.g., long paragraphs with repetitive phrasing in the Discussion and Results).
    • Grammatical issues and awkward expressions occur throughout the manuscript (e.g., "mucosal immunity in shock protein" → unclear). A professional language edit is recommended.

R: We have revised the English with the help of a native English speaker and hope that our manuscript is now correct.

Minor Suggestions

  • Reorganize the Discussionto avoid redundancy. Currently, the same proteins are repeatedly mentioned with overlapping functional speculation.
  1. We have completely reworked the discussion eliminating repetitions as much as possible

  • Specify how many total proteinswere identified before filtering.

R.: we added the number of spots corresponding to the number of proteins resolved by 2DE and added an explanation for the increase in their number (see page 11, lines 377-392)

  • Clarify whether the STRING analysis included only significantly altered proteins or the entire saliva proteome dataset.

R.: we specified that the STRING analysis was limited to the proteins included in Table 8 (see page 15, lines 531-533).

All changes in the manuscript have been highlighted in yellow

Reviewer 2 Report

Comments and Suggestions for Authors

Dear Authors,

Thank you for the opportunity to review your manuscript. Your work on developing a methodological approach for salivary proteome analysis in lung cancer screening is timely and addresses an important clinical need. The concept of utilizing saliva as a non-invasive biofluid for early cancer detection holds immense promise, and your efforts to standardize the pre-analytical phase are particularly valuable.

While I found the premise of your study compelling, I have identified several areas that require significant attention and clarification to strengthen the manuscript. My comments are structured to help you enhance the methodological robustness, interpret your findings more clearly, and refine the overall presentation.

  1. Scope and Focus: The manuscript attempts to cover both methodological development and preliminary biomarker identification. While these are related, the transition between them could be smoother. Consider whether a clearer delineation or a more focused narrative would benefit the reader.

  2. Clarity and Flow: At times, the narrative flow is a bit challenging to follow, particularly in the Results section where different experimental arms are presented. Ensuring that each experiment's objective, methods, and outcomes are clearly linked will improve readability.

  3. Statistical Rigor: While some statistical analyses are presented, the power of the preliminary patient data is limited. Acknowledging this more explicitly and discussing the implications for interpretation would be beneficial.

Specific Comments

1. Language and Style:

The language is generally academic, which is appropriate. However, there are instances where the phrasing could be more precise or where minor grammatical adjustments would enhance clarity. For example:

  • Abstract: "Currently there are no specific biomarkers for early diagnosis of lung cancer, a highly lethal tumor if diagnosed late." – While true, this could be phrased more actively, e.g., "Early diagnosis of lung cancer remains a significant challenge due to the lack of specific biomarkers, contributing to its high lethality when detected at advanced stages."

  • Introduction (Lines 79-87): The sentence structure describing the challenges of saliva use is a bit convoluted. Consider breaking down complex sentences for better readability. For instance, "first of all the presence of highly abundant proteins [13,14] that can mask the less abundant ones, which represent less than 81 2% of the proteins contained in saliva, and often include tumor biomarkers, produced in extremely small quantities by the tumor, so obviously we do not find them in large quantities in saliva." could be rephrased for conciseness and impact.

  • Throughout the manuscript: Ensure consistent use of terminology. For example, "bidimensional electrophoresis" and "two-dimensional electrophoresis (2-DE)" are both used. Stick to one.

  • Humanized Academic Tone: The current language is already quite humanized and academic, which is good. To avoid any perception of AI-generated text, focus on nuanced phrasing, perhaps introducing more active voice where appropriate, and ensuring that the conclusions drawn are always tempered with appropriate scientific caution and acknowledgment of limitations. Avoid overly generic or repetitive phrases.

2. Weak Points in Methodology:

  • Cryopreservation Stability Studies (Unstimulated Saliva - Table 1 & Figure 1):

    • Contradictory Findings: Table 1 shows a decrease in protein concentration after 6 and 12 months of cryopreservation for unstimulated saliva. However, Figure 1C (11C, 12 months) shows a remarkable increase in the number of identified spots () compared to 2 months () and 6 months (). This is a significant discrepancy. If protein concentration decreases, one would generally expect fewer, not more, detectable spots unless there's a specific explanation (e.g., protein aggregation leading to fewer but larger spots, or a change in extraction efficiency for certain proteins, or perhaps the initial protein measurement was flawed). The explanation offered (Lines 170-172: "The remarkable increase of identified spots in the sample after 12 months of cryopreservation coupled to a decrease in the measured protein content measured at the end of the cryopreservation period suggested the presence of contaminants and/or proteins in the salivary sample, capable of causing long-term denaturation of the protein components.") is speculative and needs experimental backing. How would contaminants increase the number of protein spots while decreasing the total protein concentration? This needs thorough investigation and a clearer explanation.

    • Sample Selection: For the stability studies, only 3 unstimulated saliva samples (83C, 51C, 11C) were randomly selected. While technical triplicates were performed, the biological variability among only three samples might limit the generalizability of the stability findings. A larger initial pool of samples for stability assessment would strengthen this section.

    • Protein Extraction Before Cryopreservation (Table 3 & Figure 2): The finding that protein extraction before cryopreservation decreased protein concentration and led to denaturation processes (Lines 184-187) is counterintuitive if the goal is to remove contaminants and increase stability. This result suggests that the extraction process itself might be detrimental or that the extracted proteins are more fragile in solution. This needs more detailed mechanistic explanation or further optimization.

  • Sample Size for Patient Data: The preliminary results on the patient with a lung cancer nodule (N) are based on a single patient. While acknowledged as "pilot experiments," it is crucial to emphasize that these are purely illustrative and cannot be generalized. This should be explicitly stated as a major limitation in the Discussion.

3. Weak Points in Results and Discussion:

  • Interpretation of Stability Data: The contradictory results regarding protein concentration versus spot count in unstimulated saliva (as noted above) need to be reconciled. This is a foundational aspect of your methodology, and its ambiguity weakens the subsequent findings.

  • "Matched Spots" Percentage (Table 2 & Table 4): The percentage of matched spots is reported "vs 12-month sample" (Table 2) and "vs 1 day sample" (Table 4). This reference point is arbitrary and makes cross-comparison difficult. It would be more informative to compare each time point to the "Time 0" or "immediately extracted" sample, as this directly assesses degradation over time.

  • Discussion of Biomarkers: While you list several proteins found in saliva and their known associations with lung cancer (Lines 556-690), the discussion could be more focused on how your findings contribute to this knowledge. Specifically, for the proteins identified in the pilot patient sample (PDXK, SERPB3, S100A14, AKR1A1), discuss their specific relevance in the context of salivary biomarkers and the implications of their dysregulation or exclusive presence in your single patient.

  • Lack of Quantitative Data for Biomarkers: The preliminary patient data mentions "deregulation" and "exclusive presence" of proteins. While 2D gels provide qualitative insights, quantitative data (e.g., fold changes from mass spectrometry, if available, or densitometric analysis of spots) would significantly strengthen these preliminary findings. Without quantitative data, it's hard to assess the magnitude of deregulation.

  • Future Directions: While you mention continuing the work on a larger population (Lines 697-705), the discussion could elaborate more on the specific challenges and next steps for validating these salivary biomarkers. For instance, what kind of validation cohort is envisioned? What statistical approaches will be employed to account for inter-individual variability? How will confounding factors (e.g., oral health, smoking status, other systemic diseases) be controlled for?

  • Overly Optimistic Tone: While enthusiasm is appreciated, some statements might be perceived as overly optimistic given the preliminary nature of some results. For example, "Therefore, our results, although needing further validation, suggest that the use of saliva and the proposed analytical method could favor an early identification of potential tumor biomarkers, hopefully improving clinical cancer management and patient survival" (Abstract, Lines 36-38). This is a strong claim based on limited patient data. Tempering such statements with more caution would enhance scientific credibility.

Minor Points

  • Figure Legends: Ensure all figure legends are self-explanatory and provide sufficient detail for interpretation without referring to the main text.

  • References: Double-check reference formatting and consistency.

  • Typos/Grammar: A thorough proofread for minor typos and grammatical errors is recommended. For example, "pulmunary nodules" (Line 44) should be "pulmonary nodules." "The fist one" (Line 621) should be "the first one."

I hope these comments are helpful in revising your manuscript. I look forward to seeing the improved version.

Best regards,

Comments on the Quality of English Language

The language is generally academic, which is appropriate. However, there are instances where the phrasing could be more precise or where minor grammatical adjustments would enhance clarity. For example:

  • Abstract: "Currently there are no specific biomarkers for early diagnosis of lung cancer, a highly lethal tumor if diagnosed late." – While true, this could be phrased more actively, e.g., "Early diagnosis of lung cancer remains a significant challenge due to the lack of specific biomarkers, contributing to its high lethality when detected at advanced stages."

  • Introduction (Lines 79-87): The sentence structure describing the challenges of saliva use is a bit convoluted. Consider breaking down complex sentences for better readability. For instance, "first of all the presence of highly abundant proteins [13,14] that can mask the less abundant ones, which represent less than 81 2% of the proteins contained in saliva, and often include tumor biomarkers, produced in extremely small quantities by the tumor, so obviously we do not find them in large quantities in saliva." could be rephrased for conciseness and impact.

  • Throughout the manuscript: Ensure consistent use of terminology. For example, "bidimensional electrophoresis" and "two-dimensional electrophoresis (2-DE)" are both used. Stick to one.

  • Humanized Academic Tone: The current language is already quite humanized and academic, which is good. To avoid any perception of AI-generated text, focus on nuanced phrasing, perhaps introducing more active voice where appropriate, and ensuring that the conclusions drawn are always tempered with appropriate scientific caution and acknowledgment of limitations. Avoid overly generic or repetitive phrases.

Author Response

We sincerely thank this reviewer for his/her valuable suggestions and constructive criticism. Below you will find our answers to the questions raised.

  1. Scope and Focus:The manuscript attempts to cover both methodological development and preliminary biomarker identification. While these are related, the transition between them could be smoother. Consider whether a clearer delineation or a more focused narrative would benefit the reader.

R.: We have reworked the Introduction and hope to have satisfied this Reviewer's request.

  1. Clarity and Flow:At times, the narrative flow is a bit challenging to follow, particularly in the Results section where different experimental arms are presented. Ensuring that each experiment's objective, methods, and outcomes are clearly linked will improve readability.

R.: The section of the Results was also completely revised and we hope to have better explained the different arms of our experimental protocol.

  1. Statistical Rigor:While some statistical analyses are presented, the power of the preliminary patient data is limited. Acknowledging this more explicitly and discussing the implications for interpretation would be beneficial.

R.: We have highlighted the limitations of our experiments in relation to the patient with pulmonary nodular lesion by specifying that these data are included only as an example (page 17, line 579) and that we are aware that further studies are necessary to validate them (see page 21, lines 723-733).

Specific Comments

  1. Language and Style:

The language is generally academic, which is appropriate. However, there are instances where the phrasing could be more precise or where minor grammatical adjustments would enhance clarity. For example:

  • Abstract:"Currently there are no specific biomarkers for early diagnosis of lung cancer, a highly lethal tumor if diagnosed late." – While true, this could be phrased more actively, e.g., "Early diagnosis of lung cancer remains a significant challenge due to the lack of specific biomarkers, contributing to its high lethality when detected at advanced stages."
  • Introduction (Lines 79-87):The sentence structure describing the challenges of saliva use is a bit convoluted. Consider breaking down complex sentences for better readability. For instance, "first of all the presence of highly abundant proteins [13,14] that can mask the less abundant ones, which represent less than 81 2% of the proteins contained in saliva, and often include tumor biomarkers, produced in extremely small quantities by the tumor, so obviously we do not find them in large quantities in saliva." could be rephrased for conciseness and impact.
  • Throughout the manuscript:Ensure consistent use of terminology. For example, "bidimensional electrophoresis" and "two-dimensional electrophoresis (2-DE)" are both used. Stick to one.
  • Humanized Academic Tone:The current language is already quite humanized and academic, which is good. To avoid any perception of AI-generated text, focus on nuanced phrasing, perhaps introducing more active voice where appropriate, and ensuring that the conclusions drawn are always tempered with appropriate scientific caution and acknowledgment of limitations. Avoid overly generic or repetitive phrases.

R.: We took all these suggestions into consideration and thoroughly revised the manuscript, including with the help of an English speaker. We hope our article has improved in this respect as well.

  1. Weak Points in Methodology:
  • Cryopreservation Stability Studies (Unstimulated Saliva - Table 1 & Figure 1):
    • Contradictory Findings:Table 1 shows a decrease in protein concentration after 6 and 12 months of cryopreservation for unstimulated saliva. However, Figure 1C (11C, 12 months) shows a remarkable increase in the number of identified spots (541±66) compared to 2 months (173±28) and 6 months (102±16). This is a significant discrepancy. If protein concentration decreases, one would generally expect fewer, not more, detectable spots unless there's a specific explanation (e.g., protein aggregation leading to fewer but larger spots, or a change in extraction efficiency for certain proteins, or perhaps the initial protein measurement was flawed). The explanation offered (Lines 170-172: "The remarkable increase of identified spots in the sample after 12 months of cryopreservation coupled to a decrease in the measured protein content measured at the end of the cryopreservation period suggested the presence of contaminants and/or proteins in the salivary sample, capable of causing long-term denaturation of the protein components.") is speculative and needs experimental backing. How would contaminants increase the number of protein spots while decreasing the total protein concentration? This needs thorough investigation and a clearer explanation.

R.: We attempted to explain these contradictory results on page 19 (lines 644-654). Indeed, both prolonged cryopreservation can alter protein composition, increasing the activity of contaminants and/or the creation of aggregates that escape protein quantification, while also favoring the formation of peptide fragments that increase the number of spots resolved by 2DE. We based our experimental protocol on articles demonstrating the potential degradation of proteins depending on sample storage conditions (see references. Xx and 42).

  • Sample Selection:For the stability studies, only 3 unstimulated saliva samples (83C, 51C, 11C) were randomly selected. While technical triplicates were performed, the biological variability among only three samples might limit the generalizability of the stability findings. A larger initial pool of samples for stability assessment would strengthen this section.

R.: We have increased the number of samples analyzed for protein concentration measurement (see pages 4 and 7).

  • Protein Extraction Before Cryopreservation (Table 3 & Figure 2):The finding that protein extraction before cryopreservation decreased protein concentration and led to denaturation processes (Lines 184-187) is counterintuitive if the goal is to remove contaminants and increase stability. This result suggests that the extraction process itself might be detrimental or that the extracted proteins are more fragile in solution. This needs more detailed mechanistic explanation or further optimization.

R.: Our results show that extraction before sample storage cannot completely prevent partial protein degradation, although it reduces the presence of potential contaminants. Indeed, the number of spots after prolonged cryopreservation periods decreased compared to the results obtained in samples where proteins were extracted after cryopreservation. We acknowledge that these results are contradictory and require further investigation (page 7, lines 232-233). Therefore, since protein extraction before cryopreservation did not provide any benefit in terms of contaminants or protein degradation, we decided to perform the extraction after cryopreservation, as this ensures simpler sample handling.

  • Sample Size for Patient Data:The preliminary results on the patient with a lung cancer nodule (N) are based on a single patient. While acknowledged as "pilot experiments," it is crucial to emphasize that these are purely illustrative and cannot be generalized. This should be explicitly stated as a major limitation in the Discussion.

R.: We agree on this aspect and have highlighted that these data are reported only as examples and are preliminary, requiring further validation (see page 21, lines 723-733).

  1. Weak Points in Results and Discussion:
  • Interpretation of Stability Data:The contradictory results regarding protein concentration versus spot count in unstimulated saliva (as noted above) need to be reconciled. This is a foundational aspect of your methodology, and its ambiguity weakens the subsequent findings.

R.: As reported above, we have attempted to explain this aspect in the Results section, on page 5, lines 169-175, page 7, lines 232-233 and in the Discussion, page 19 lines 644-654, citing also two references.

  • "Matched Spots" Percentage (Table 2 & Table 4):The percentage of matched spots is reported "vs 12-month sample" (Table 2) and "vs 1 day sample" (Table 4). This reference point is arbitrary and makes cross-comparison difficult. It would be more informative to compare each time point to the "Time 0" or "immediately extracted" sample, as this directly assesses degradation over time.

R.: Unfortunately, we did not perform 2DE at time 0 for these samples; however, the increase in spots observed after 12 months of cryopreservation is evident and consistent with the formation of peptide fragments that increased the number of electrophoretically resolved spots.

  • Discussion of Biomarkers:While you list several proteins found in saliva and their known associations with lung cancer (Lines 556-690), the discussion could be more focused on how your findings contribute to this knowledge. Specifically, for the proteins identified in the pilot patient sample (PDXK, SERPB3, S100A14, AKR1A1), discuss their specific relevance in the context of salivary biomarkers and the implications of their dysregulation or exclusive presence in your single patient.

R.: We fully agree with this suggestion. Therefore, we focused a substantial portion of the discussion on the points raised by this reviewer and hope we have satisfied his request.

  • Lack of Quantitative Data for Biomarkers:The preliminary patient data mentions "deregulation" and "exclusive presence" of proteins. While 2D gels provide qualitative insights, quantitative data (e.g., fold changes from mass spectrometry, if available, or densitometric analysis of spots) would significantly strengthen these preliminary findings. Without quantitative data, it's hard to assess the magnitude of deregulation.

R.: We introduced several quantitative data on the obtained results, such as 1) the magnification of the spots related to proteins differentially expressed in stimulated saliva samples compared to those in unstimulated samples, coupled with histograms revealing differences in their intensity (see the new figure 7); 2) the significant P value calculated during MS analysis to confirm these differences and reported in the last column on the right in Table 8; 3) we added Figure S1 to the Supplementary Material. It reports a scatter plot analysis that demonstrates the validity of our results in the choice of proteins identified by MS as common in the two types of saliva samples (for an explanation see page 11, lines 386-392).

  • Future Directions:While you mention continuing the work on a larger population (Lines 697-705), the discussion could elaborate more on the specific challenges and next steps for validating these salivary biomarkers. For instance, what kind of validation cohort is envisioned? What statistical approaches will be employed to account for inter-individual variability? How will confounding factors (e.g., oral health, smoking status, other systemic diseases) be controlled for?

R.: We thank the reviewer for this suggestion and have attempted to explain how the results will be organized in our future studies in the Conclusions (p. 24). Obviously, the evaluations are still in their early stages, and therefore the statistical evaluations, the division of patients into cohorts based on both the tumor pathology detected in the lung and other health-related factors of the studied patients, will be better defined once we have more results.

  • Overly Optimistic Tone:While enthusiasm is appreciated, some statements might be perceived as overly optimistic given the preliminary nature of some results. For example, "Therefore, our results, although needing further validation, suggest that the use of saliva and the proposed analytical method could favor an early identification of potential tumor biomarkers, hopefully improving clinical cancer management and patient survival" (Abstract, Lines 36-38). This is a strong claim based on limited patient data. Tempering such statements with more caution would enhance scientific credibility.

R.: We also thank the reviewer for this kind suggestion. We have modified the text of the Abstract, Introduction and Discussion, limiting our enthusiasm. We hope that this will be a good fit for the paper.

Minor Points

  • Figure Legends:Ensure all figure legends are self-explanatory and provide sufficient detail for interpretation without referring to the main text.

R.: we revised and corrected all legends.

  • References:Double-check reference formatting and consistency.

R.: we checked the references and added many others to better document our findings

  • Typos/Grammar:A thorough proofread for minor typos and grammatical errors is recommended. For example, "pulmunary nodules" (Line 44) should be "pulmonary nodules." "The fist one" (Line 621) should be "the first one."

I hope these comments are helpful in revising your manuscript. I look forward to seeing the improved version.

R.: We thank the reviewer for his/her patience. We hope the errors have been corrected and no new ones have been added.

Comments on the Quality of English Language

The language is generally academic, which is appropriate. However, there are instances where the phrasing could be more precise or where minor grammatical adjustments would enhance clarity. For example:

  • Abstract:"Currently there are no specific biomarkers for early diagnosis of lung cancer, a highly lethal tumor if diagnosed late." – While true, this could be phrased more actively, e.g., "Early diagnosis of lung cancer remains a significant challenge due to the lack of specific biomarkers, contributing to its high lethality when detected at advanced stages."
  • Introduction (Lines 79-87):The sentence structure describing the challenges of saliva use is a bit convoluted. Consider breaking down complex sentences for better readability. For instance, "first of all the presence of highly abundant proteins [13,14] that can mask the less abundant ones, which represent less than 81 2% of the proteins contained in saliva, and often include tumor biomarkers, produced in extremely small quantities by the tumor, so obviously we do not find them in large quantities in saliva." could be rephrased for conciseness and impact.
  • Throughout the manuscript:Ensure consistent use of terminology. For example, "bidimensional electrophoresis" and "two-dimensional electrophoresis (2-DE)" are both used. Stick to one.
  • Humanized Academic Tone:The current language is already quite humanized and academic, which is good. To avoid any perception of AI-generated text, focus on nuanced phrasing, perhaps introducing more active voice where appropriate, and ensuring that the conclusions drawn are always tempered with appropriate scientific caution and acknowledgment of limitations. Avoid overly generic or repetitive phrases.

R.: We've completely reworked all sections of our manuscript to improve clarity and avoid repetition. We've tried to check the spelling of words and maintain acronyms throughout the manuscript, once we've established their meaning the first time they appear in the text.

We've done our best to improve the quality of our article, including with the help of a native speaker, and we hope the reviewer will agree.

All changes in the manuscript have been highlighted in yellow

Round 2

Reviewer 1 Report

Comments and Suggestions for Authors

Authors answered all my questions. I agree to publish this version. 

Reviewer 2 Report

Comments and Suggestions for Authors

Dear Authors,

I appreciate the effort and diligence you have shown in revising your manuscript. Your detailed responses and the corresponding changes have effectively addressed all of my previous concerns. The improvements in clarity, data presentation, and discussion have notably enhanced the overall quality of the work.